

# A Deep-Learning Hybrid-Predictive-Modeling Approach for
# Estimating Evapotranspiration and Ecosystem Respiration
Jiancong Chen[1], Baptiste Dafflon[2], Anh Phuong Tran[2, 3], Nicola Falco[2], and Susan S. Hubbard[2]
[1]Department of Civil and Environmental Engineering, University of California, Berkeley, CA, USA, [2]Earth and
Environmental Sciences Area, Lawrence Berkeley National Laboratory, Berkeley, CA, USA, [3]Depatment of Water
Resources Engineering and Technology, Water Resources Institute, 8, Phao Dai Lang, Dong Da, Hanoi, Vietnam
**Abstract:** Gradual changes in meteorological forcings (such as temperature and precipitation) are reshaping
vulnerable ecosystems, leading to uncertain effects on ecosystem dynamics, including water and carbon fluxes.
Estimating evapotranspiration (ET) and ecosystem respiration ($R_{ECO}$) is essential for analyzing the effect of climate
change on ecosystem behavior. To obtain a better understanding of these processes, we need to improve our estimation
of water and carbon fluxes over space and time, which is difficult within ecosystems where we have only sparse data.
In this study, we developed a hybrid predictive modeling approach (HPM) that integrates eddy covariance
measurements, physically-based model simulation results, meteorological forcings, and remote sensing datasets to
estimate evapotranspiration (ET) and ecosystem respiration ($R_{ECO}$) in high space-time resolution. HPM relies on a
deep learning algorithm-long short term memory (LSTM)-as well as direct measurements or outputs from physically-
based models.  We tested and validated HPM estimation results at sites within various mountainous regions, given
their importance for water resources, their vulnerability to climate change, and the recognized difficulties in estimating
ET and $R_{ECO}$ in mountainous regions. We benchmarked estimates of ET and  $R_{ECO}$ obtained from the HPM method
against measurements made at FLUXNET stations and outputs from the Community Land Model (CLM) at Rocky
Mountain SNOTEL stations. At the mountainous East River Watershed site in the Upper Colorado River Basin, we
explored how ET and $R_{ECO}$ dynamics estimated from the new HPM approach vary with different vegetation and
meteorological forcings. The results of this study indicate that HPM is capable of identifying complicated interactions
among meteorological forcings, ET, and $R_{ECO}$ variables, as well as providing reliable estimation of ET and $R_{ECO}$
across relevant spatiotemporal scales. With HPM estimation of ET and $R_{ECO}$ at the East River Watershed, we found
that abiotic factors of temperature and radiation predominantly explained ET spatial variability; whereas $R_{ECO}$
variability was largely controlled by biotic factors, such as vegetation type. In general, our study demonstrated that
the HPM approach can circumvent the typical lack of spatiotemporally dense data needed to estimate ET and $R_{ECO}$
over space and time, as well as the parametric and structural uncertainty inherent in mechanistic models. While the
current limitations of the HPM approach are driven by the temporal and spatial resolution of available datasets (such
as NDVI), ongoing advances in remote sensing are expected to further improve accuracy and resolution of ET and
$R_{ECO}$ estimation using HPM.
**1. Introduction:**
Evapotranspiration (ET) and ecosystem respiration ($R_{ECO}$) are key components of ecosystem water and
carbon cycles. ET is an important link between the water and energy cycles: dynamic changes in ET can affect
precipitation, soil moisture, and surface temperature, leading to uncertain feedbacks in the environment (Jung et al.,
2010; Seneviratne et al., 2006; Teuling et al., 2013). Thus, quantifying ET is particularly essential for improving our





understanding of water and energy interactions and watershed response to abrupt and gradual changes in climate,
which is critical for water resources management, agriculture, and other societal benefits (Anderson et al., 2012; Jung
et al., 2010; Rungee et al., 2019; Viviroli et al., 2007; Viviroli and Weingartner, 2008). $R_{ECO}$ describes the sum of
autotrophic respiration and respiration by heterotrophic microorganisms  in a specific ecosystem and plays a vital role
in the response of terrestrial ecosystem to global change (Jung et al., 2017; Reichstein et al., 2005; Xu et al., 2004).
As long term exchanges in $R_{ECO}$ have pivotal influences over the climate system (Cox et al., 2000;  Gao et al., 2017;
IPCC, 2019; Suleau et al., 2011), approaches are needed to estimate and monitor $R_{ECO}$ over relevant spatiotemporal
scales. As described below, there are many different strategies for measuring and estimating ET and $R_{ECO}$, each of
which has advantages and limitations. The motivation for this study is the recognition that current methods cannot
provide ET and $R_{ECO}$ at space and time scales needed to improve prediction of changing terrestrial system behavior,
particularly in challenging mountainous watersheds.

Several ground-based approaches have been used to provide *in situ* estimates or measurements of ET and

$R_{ECO}$. Ground based flux chambers capture and measure trace gases emitted from the land surface, which can be used
to estimate ET and $R_{ECO}$ (Livingston and Hutchinson, 1995; Pumpanen et al., 2004). However, the microclimate of
the environment is affected by the chamber, and the laborious acquisition process and small chamber size typically
lead to information with coarse spatiotemporal resolution (Baldocchi, 2014). The eddy covariance method uses a tower
with installed instruments to autonomously measure fluxes of trace gases between ecosystem and atmosphere
(Baldocchi, 2014; Wilson et al., 2001). The covariance between the vertical velocity and mixing ratios of the target
scalar is computed to obtain the fluxes of carbon, water vapor, and other trace gases emitted from the land surface. ET
is then calculated from the latent heat flux, and $R_{ECO}$ is calculated from the net carbon fluxes using night-time or
daytime partitioning approaches (van Gorsel et al., 2009; Lasslop et al., 2010; Reichstein et al., 2005). The spatial
footprint of obtained fluxes is on the order of hundreds of meters, and the temporal resolution of the measurements
range from hours to decades (Wilson et al., 2001). Such *in situ* measurements of fluxes have been integrated into the
global network of AmeriFlux (http://ameriflux.lbl.gov/) and FLUXNET (https://FLUXNET.fluxdata.org/), where
such data have strongly supported scientists in process understanding and model development. Given the cost, efforts,
and power required to install and maintain a flux tower, eddy covariance towers are typically sparse relative to the
scale of study sites used to address ecosystem questions. Additionally, the location of a flux tower within a watershed
greatly influences measurement representativeness. For example, eddy covariance towers are usually installed at
valley bottoms of mountainous watersheds (Strachan et al., 2016), and estimates obtained there may not be
representative of fluxes across a range of elevations or slope aspects within the watershed. The limited number of
towers and their limited ability to sample different portions of a watershed thus limit the usefulness of flux towers for
estimating ET and $R_{ECO}$ in high resolution over space and time.

Physically-based models, which numerically represent land-surface energy and water balance, have also been

used to estimate ET and $R_{ECO}$ (Tran et al., 2019; Williams et al., 2009). These physically-based models solve physical
equations to simulate the exchanges of energy, heat, water and carbon across atmosphere-canopy-soil compartments.
Examples include the Community Land Model (CLM, Oleson et al., 2013). Performance of these models depend on



the accuracy of inputs and parameters, such as soil type and leaf area index, which can be difficult to obtain at
sufficiently high spatiotemporal resolution. The lack of measurements to infer parameters needed for models often
leads to large discrepancies between model-based and flux-tower-based ET and $R_{ECO}$ estimates. Conceptual model
uncertainty inherent in mechanistic models can also lead to ET and $R_{ECO}$ estimation uncertainty and errors. For
example, Keenan et al. (2019) suggested that current terrestrial carbon cycle models neglect inhibition of leaf
respiration that occurs during daytime, which can result in a bias of up to 25%. These conceptual uncertainties, in
addition to data sparseness and data uncertainty, further limit the applicability of physically-based models to estimate
ET and $R_{ECO}$ at high spatiotemporal scales. Semi-analytical formulations based on combinations of meteorological
and empirical parameters provide a reference condition for the water and energy balance. Examples used to estimate
potential ET include the Budyko framework and its extensions (Budyko, 1961; Greve et al., 2015; Zhang et al., 2008);
the Penman-Monteith's equation (Allen et al., 1998), and the Priestley-Taylor equation (Priestley and Taylor, 1972).
Actual ET can then be approximated by multiplying a coefficient associated with water deficit (De Bruin, 1983;
Williams & Albertson, 2004). However, even with these empirical formulations many attributes are still difficult to
obtain globally at high temporal scales, such as water-vapor deficit, leaf area index, and aerodynamic conductance of
different plants.
Remote sensing products, such as Landsat imagery (Irons et al., 2012) and the moderate-resolution imaging
spectroradiometer (MODIS, NASA. 2008), have also been integrated to estimate ET and $R_{ECO}$ with empirical,
statistical, or semi-physical relations (Abatzoglou et al., 2014; Daggers et al., 2018; Mohanty et al., 2017; Paca et al.,
2019). Due to the high spatial coverage of remote sensing products, global-scale estimates of ET and $R_{ECO}$ have
become feasible. For example, Ryu et al. (2011) proposed the Breathing Earth System Simulator approach, which
integrates mechanistic models and MODIS data to quantify ET and GPP with a spatial resolution of 1-5 km and a
temporal resolution of 8 days. Ai et al. (2018) extracted enhanced vegetation index, fraction of absorbed
photosynthetically active radiation, and leaf area index from the MODIS dataset—and used the rate-temperature curve
and strong correlations between terrestrial carbon exchange and temperature to estimate $R_{ECO}$ at 1 km spatial
resolution and 8-day temporal resolution. Ma et al. (2018) developed a data fusion scheme that fused Landsat-like-
scale datasets and MODIS data to estimate ET and irrigation water efficiency at a spatial scale of ~100 meters.
However, even though remote sensing data cover large areas of the earth surface, they typically do not provide
information over both high spatial and temporal resolution, and are also subject to cloudy conditions. For example,
Landsat has average return periods of 16 days with a spatial resolution of 30 m (visible and near-infrared), whereas
MODIS has 1-2 days temporal resolution with a 250 m or 1 km spatial resolution depending on the sensors. These
resolutions are typically too coarse to enable exploration of how aspects such as plant phenology, snowmelt, and
rainfall impact integrated ecosystem water and energy dynamics.
Combining machine-learning models with remote sensing products and meteorological inputs offers another
option for large-scale estimation of ET and $R_{ECO}$. Remotely sensed data are good proxies for plant productivity and
can be easily implemented into machine-learning models for ET and $R_{ECO}$ estimation, such as for an enhanced
vegetation index, land surface water index and NDVI (Gao et al., 2015; Jägermeyr et al., 2014; Migliavacca et al.,





2015). Li and Xiao (2019) developed a data-driven model for gross primary production at a spatial and temporal
resolution of 0.05° and 8 days using MODIS and meterological reanalysis data. Berryman et al. (2018) demonstrated
a Random Forest model to predict growing season soil respiration from subalpine forests in the Southern Rocky
Mountains ecoregion. Jung et al. (2009) developed a model tree ensemble approach to upscale FLUXNET data, where
they have successfully estimated ET and GPP. Other methods have used support vector machines, artificial neural
networks, random forest, and piecewise regression (Bodesheim et al., 2018; Metzger et al., 2013; Xiao et al., 2014;
Xu et al., 2018). These models were trained with ground-measured flux observations and other variables, and then
applied to estimate ET over continental or global scales with remote sensing and meteorological inputs. Some of the
most important inputs include the enhanced vegetation index, aridity index, temperature, and precipitation. However,
the spatiotemporal resolution of these approaches is constrained by the resolution of remote sensing products and
meteorological inputs. Additionally, parameters such as leaf area index, cloudiness, and the vegetation types required
by those models may not be available at the required resolution, accuracy or location. For example, in systems that
have significant elevation gradients, errors may result when valley-based FLUXNET data are used for training and
then applied to hillslope or ridge ET and $R_{ECO}$ estimation
Development of hybrid models that link direct measurements and/or interpretable mechanistic models with
data-driven methods can benefit ET and $R_{ECO}$ estimation (Reichstein et al., 2019). While remote sensing data that
cover large regions provide promise for informing models, quantitative interpretation of these data needed for input
into mechanistic models is still challenging (Reichstein et al., 2019). Physically-based models can provide estimates
of ET and $R_{ECO}$, but the estimate error can be high, owing to parametric, structural, and conceptual uncertainties as
described above. Hybrid data-driven frameworks are potentially advantageous because they enable the integration of
remote sensing datasets, meteorological forcings, and mechanistic model outputs of ET and $R_{ECO}$ into one model.
Machine-learning approaches are then applied to extract the spatiotemporal patterns for ET and $R_{ECO}$ prediction.
Hybrid models can utilize the high spatial coverage of remote sensing data (e.g., 30 m of Landsat) and high temporal
resolution of direct measurement from flux towers or simulation results from mechanistic models (e.g., daily or hourly
scales), thus providing alternative approaches for next-stage, more accurate estimation of ET and $R_{ECO}$ at greater
spatial and finer temporal scales—and enhancing our process understanding of water and carbon cycling under climate
change.
In this study, we developed a hybrid predictive modeling approach (HPM) to better estimate ET and $R_{ECO}$
over space and time with easily acquired meteorological data (i.e., air temperature, precipitation and radiation) and
remote sensing products (i.e., NDVI). HPM is hybrid as it can use deep learning models to integrate direct
measurements from flux towers and physically-based model results (e.g., CLM) with meteorological and remote
sensing inputs to capture the complex physical interactions within the watershed ecosystem. After development, we
validated HPM performance with the FLUXNET dataset and benchmarked the CLM model at select sites. We then
used the HPM for ET and $R_{ECO}$ estimation at the mountainous East River Watershed in CO and investigated how
small-scale heterogeneity influences ET and $R_{ECO}$ dynamics.





The remainder of this paper is organized as follows. Section 2 mainly describes the sites considered in this
study and how data were acquired and processed. Section 3 presents the methodology of the HPM approach, followed
by the results of various use cases presented in Section 4. Discussion and conclusion are provided in Sections 5 and
6, respectively.
**2. Site Information, Data Acquisition and Processing**
We selected various sites to develop and validate our approaches. We focused on mountainous watersheds
because they provide significant water resources to the world (Viviroli et al., 2007), but also included sites to test
HPM's capabilities under different climate and vegetation conditions. Mountainous watersheds are very sensitive to
changes in temperature and precipitation patterns, which can significantly threaten downgradient water resources and
associated societal benefits (Breshears et al., 2005; Ernakovich et al., 2014; Immerzeel et al., 2019). As mountainous
regions are extremely important for regional and global assessment and management of water resources and carbon
storage and emission (Knowles et al., 2015; Schimel et al., 2002), accurate estimation of ET and $R_{ECO}$ in these regions
is critical, though challenging due to complex heterogeneity and complicated interactions among the hydrosphere,
biosphere and the atmosphere (Pelletier et al., 2018; Speckman et al., 2015). Thus, we focused on estimating ET and
$R_{ECO}$ at various sites along the Rocky Mountains, including the East River Watershed  (Hubbard et al., 2018) of the
Upper Colorado River Basin.
**2.1 FLUXNET Stations and Ecoregions**
Eight FLUXNET stations were selected for this study (Table 1 and Figure 1), which cover a wide range of
climate and vegetation types. These stations have elevations from 129 m (US-Var) to 3050 m (US-NR1), mean annual
air temperature from 1.5℃ (US-NR1) to 17.92℃ (US-SRM), and mean annual precipitation from 320 mm (US-Whs)
to 800 mm (US-NR1). These FLUXNET stations also cover a wide range of vegetation types (i.e., evergreen forest,
deciduous forest, and shrublands). As indicated by Hargrove et al. (2003), FLUXNET stations provide a good
representation of different ecoregions, which are areas that display recurring patterns of similar combinations of soil
and landform characteristics (Omernik, 2004). Omernik & Griffith. (2014) delineated the boundaries of ecoregions
through pattern analysis that consider the spatial correlation of both physical and biological factors (i.e., soils,
physiography, vegetation, land use, geology and hydrology) in a hierarchical level. FLUXNET stations considered in
this study mainly locate in 4 unique ecoregions (Table 1). As is described below, we developed local-scale (i.e., point
scale) HPM that are representative for different ecoregions using data provided at these FLUXNET stations to estimate
ET and $R_{ECO}$, and validated the HPM estimates with measurements from stations within the same ecoregion.
**2.2 SNOTEL Stations**
For reasons described below, we performed a deeper exploration within one of the mountainous watershed
sites (the East River Watershed of the Upper Colorado River Basin), which is located in the "western cordillera"
ecoregion. At this site, we utilized meteorological forcings data from three snow telemetry (SNOTEL) stations. These
sites include the Butte (ER-BT, id: 380), Porphyry Creek (ER-PK, id: 701) and Schofield Pass (ER-SP, id: 737) sites.
A CLM model was developed at these SNOTEL stations that provides physically-model-based ET estimation (Tran





et al., 2019). Table 1 summarizes the SNOTEL stations used in this study and the corresponding climate characteristics.
Figure 1 shows the geographical locations of FLUXNET and SNOTEL stations selected in this study.
**Table 1. Summary of FLUXNET stations and SNOTEL stations information. * denotes SNOTEL stations and all others**
**are FLUXNET stations. Dfc, Bsk, Csa represent subarctic or boreal climates, semi-arid climate, Mediterranean hot summer**
**climates, respectively. ENF, DBF, WSA, GRA, and OSH represent evergreen needleleaf forest, deciduous broadleaf forests,**
**woody savannas, grasslands, open shrubland, respectively.**

| Site ID | Site Name | Latitude, Longitude | Elevation (m) | Mean Annual temperature (°C) | Mean Annual Precipitation (m) | Climate Koeppen | Vegetation IGBP | Ecoregions (Level II) | Period of Record |
|---|---|---|---|---|---|---|---|---|---|
| US-NR1 | Niwot Ridge | (40.0329, -105.5464) | 3050 | 1.5 | 800 | Dfc | ENF | Western Cordillera | 2000-2014 |
| CA-Oas | Saskatchewan-Aspen | (53.6289, -106.1978) | 530 | 0.34 | 428.53 | Dfc | DBF | Boreal Plain | 1997-2010 |
| CA-Obs | Saskatchewan-Black Spruce | (53.9872, -105.1178) | 628.94 | 0.79 | 405.6 | Dfc | ENF | Boreal Plain | 1999-2010 |
| US-SRM | Santa Rita Mesquite | (31.8214, -110.8661) | 1120 | 17.92 | 380 | Bsk | WSA | Western Sierra Madre Piedmont | 2005-2015 |
| US-Ton | Tonzi Ranch | (38.4316, -120.9660) | 177 | 15.8 | 559 | Csa | WSA | Mediterranean California | 2002-2015 |
| US-Var | Vaira Ranch-lone | (38.4133, -120.9507) | 129 | 15.8 | 559 | Csa | GRA | Mediterranean California | 2002-2015 |
| US-Whs | Walnut Gulch Lucky Hills Shrub | (31.7438, -110.0522) | 1370 | 17.6 | 320 | Bsk | OSH | Western Sierra Madre Piedmont | 2008-2015 |
| US-Wkg | Walnut Gulch Kendall Grasslands | (31.7365, -109.9419) | 1531 | 15.64 | 407 | Bsk | GRA | Western Sierra Madre Piedmont | 2005-2015 |
| ER-BT* | East River-Butte | (38.894, -106.945) | 3096 | 2.38 | 821 | Dfc | N/A | Western Cordillera | 1995-2017 |
| ER-SP* | East River-Schofield Pass | (39.02, -107.05) | 3261 | 2.46 | 1064 | Dfc | N/A | Western Cordillera | 1995-2017 |
| ER-PK* | East River-Porphyry Creek | (38.49, -106.34) | 3280 | 1.97 | 574 | Dfc | N/A | Western Cordillera | 1995-2017 |


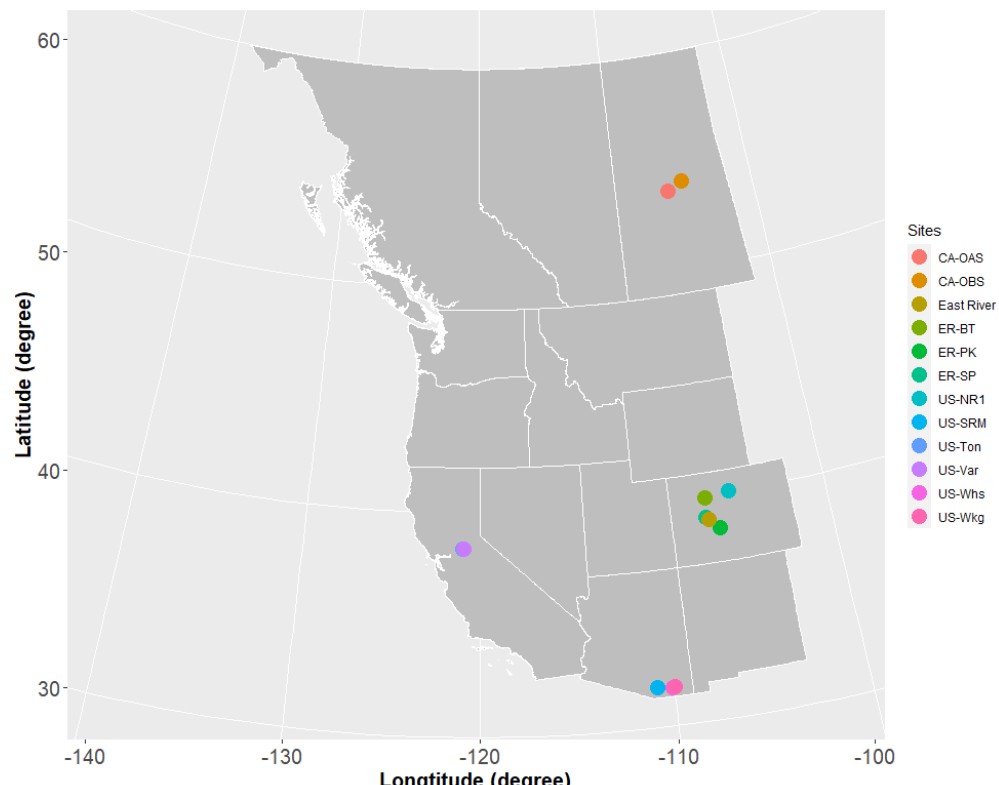


**Figure 1. Location of sites considered in this study. Note: US-Ton and US-Var; US-Whs and US-Wkg are at the same locations. East River Watershed is located next to ER-BT. The white lines delineate Western US states and Canadian provinces.**

**2.3 East River Watershed and Previous Analyses**
Data from the East River Watershed were used to explore how ET and $R_{ECO}$ dynamics estimated from the
developed HPM vary with different vegetation and meteorological forcings. The East River Watershed is located
northeast of the town of Crested Butte, Colorado. This watershed has an average elevation of 3266 m, with significant
gradients in topography, hydrology, geomorphology, vegetation, and weather. The watershed has a mean annual
temperature around 0℃ , with an average of 1200 mm yr$^{-1}$ total precipitation (Hubbard et al., 2018). Consisting of
montane, subalpine, and alpine life zones, each with distinctive vegetation biodiversity, the East River Watershed is a
testbed for the US Department of Energy Watershed Function Scientific Focus Area Project, led by the Lawrence
Berkeley National Laboratory (LBNL; Hubbard et al., 2018). The project has acquired a range of datasets, including
hydrological, biogeochemical, remote sensing, and geophysical datasets.
Recently completed studies at the East River Watershed were used in this study to inform HPM and to assess
the results. For example, physically-model-based estimations of ET at this site (Tran et al., 2019) were used herein for
HPM development and validation. Falco et al. (2019) used machine-learning-based remote sensing methods to





characterize the spatial distribution of vegetation types, slopes, and aspects within a hillslope at the East River
Watershed, which were used with obtained HPM estimates to explore how small-scale vegetation heterogeneity
influences ET and $R_{ECO}$ dynamics. To perform this assessment, we computed the spatial distribution of vegetation
types at watershed scale, based on Falco et al. (2019), and selected 16 locations within the East River Watershed
having different vegetation types and slope aspects. These 16 locations were chosen at a level to be distinguishable
by Landsat images and maintain the same vegetation type (given a spatial resolution of 30 m), and also possess small-
scale heterogeneity. A summary of the locations is presented in Table 2; the spatial distribution of the locations is
shown in Figure 2.
**Table 2: Location and vegetation types of East River Watershed sampling points (Figure 2)**

| Easting (m) | Northing (m) | Vegetation Type | Aspect | Elevation (m) |
|---|---|---|---|---|
| 327085 | 4309878 | Deciduous Forest | South | 2983 |
| 326288 | 4312504 | Deciduous Forest | South | 3177 |
| 330012 | 4313132 | Deciduous Forest | North | 3108 |
| 326854 | 4313192 | Deciduous Forest | South | 3098 |
| 328246 | 4312832 | Meadow | South | 3095 |
| 327010 | 4315059 | Meadow | South | 2790 |
| 328738 | 4306139 | Meadow | North | 2890 |
| 334270 | 4309465 | Meadow | North | 2929 |
| 333406.5 | 4308340 | Riparian Shrubland | South | 2760 |
| 327846 | 4312497 | Riparian Shrubland | South | 2723 |
| 334641 | 4305632 | Riparian Shrubland | North | 2740 |
| 330760 | 4310097 | Riparian Shrubland | South | 2855 |
| 329573 | 4314569 | Evergreen Forest | South | 3026 |
| 333106 | 4307313 | Evergreen Forest | North | 3102 |
| 325056 | 4310456 | Evergreen Forest | South | 2961 |
| 335141 | 4309614 | Evergreen Forest | North | 3131 |


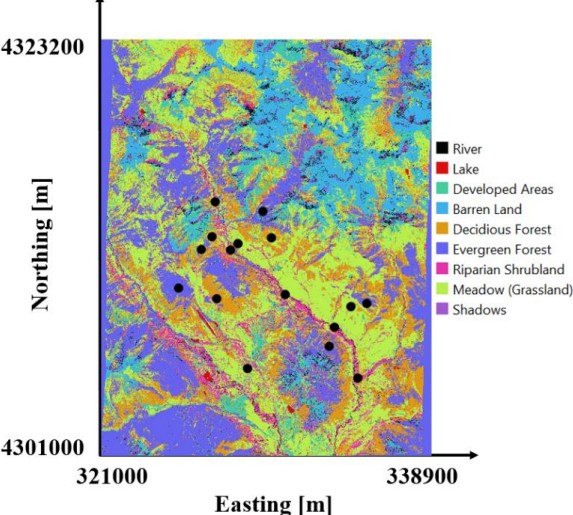


**Figure 2: Vegetation classification of the East River, CO Watershed from Falco et al. (2019). East River sites selected in**
**this study are denoted by black circles.**
**2.4 Data Collection and Processing**





To enhance transferability of the developed HPM strategy to less intensively characterized watersheds, we
selected only "easy to measure" or "widely available" attributes, such as precipitation, temperature, radiation and
NDVI, as inputs to the HTM model. The data sources used for these inputs include FLUXNET data
(https://fluxnet.fluxdata.org/), SNOTEL data (https://www.wcc.nrcs.usda.gov/snow/) and developed CLM model
(Tran et al., 2019) at SNOTEL stations, DAYMET meteorological inputs (Thornton et al., 2017) and remote sensing
data from Landsat imageries (Irons et al., 2012).
A variety of measured data and model outputs were used to train and validate HPM. We obtained daily
meteorological data, including air temperature, precipitation, radiation, ET, and $R_{ECO}$ data, from the FLUXNET
database at the selected FLUXNET sites. The pipeline of data processing for FLUXNET dataset is provided at
https://FLUXNET.fluxdata.org/. ET data for US-NR1 were cleaned following the procedures presented in Rungee et
al. (2019). The meteorological data were used as inputs for HPM development, and ET and $R_{ECO}$ data from these sites
were used for HPM validation. At the three selected SNOTEL stations, we obtained air temperature, precipitation, and
snow-water-equivalent data from the SNOTEL database. Air temperature data at these three SNOTEL stations were
processed following Oyler et al. (2015), given potential systematic artifacts. Snow-water-equivalent data are not easily
acquired, and thus were not considered as inputs for HPM. However, a categorical variable was constructed to
assimilate information regarding snow (Section 3.2.1). CLM models were generated following Tran et al. (2019) for
the SNOTEL stations and US-NR1 to assess the spatiotemporal variability of ET at the East River Watershed and for
training and validating HPM (Section 4.3). The DAYMET dataset (Thornton et al., 2017) provided gridded daily
weather-forcings-attribute estimates at a 1 km spatial resolution. We obtained the incident radiation data from
DAYMET at the SNOTEL stations as inputs for HPM. For the East River Watershed sites, meteorological forcings
data, including air temperature, precipitation and radiation, were also obtained from DAYMET. The low spatial
resolution of DAYMET data introduces uncertainty in HPM estimation of ET and $R_{ECO}$, which will be discussed in
the following sections. We calculated the NDVI time series from the red band (RED) and near-infrared band (NIR)
from Landsat 5, 7, and 8 images following Equation 1 at all selected FLUXNET sites, SNOTEL stations, and East
River Watershed sites at a spatial scale of 30 m.
$$NDVI = \frac{NIR - RED}{NIR + RED} \qquad\qquad (1)$$

Since cloud conditions can severely decrease data quality, we used the cloud-scoring algorithm provided in the Google
Earth Engine to mask clouds in all retrieved data, only selecting the ones that had a simple cloud score below 20 to
ensure data quality. Given the different calibration sensors used in Landsat 5, 7, and 8, we also followed the processes
described in Homer et al. (2015) and Vogelmann et al. (2001) to keep NDVI computations consistent over time.
Landsat satellites have a return period of 16 days, and thus we performed a reconstruction of NDVI time series to
obtain daily scale time data (Section 3.2.2).
**3. Hybrid Predictive Modeling Framework**





In this section, we illustrate the steps for building an HPM model for ET and $R_{ECO}$ estimation over time and
space. Figure 3 presents the general framework of HPM, which includes modules for data preprocessing, model
development, model validation, and predictive modeling.

**3.1 Model Framework**

HPM establishes relationships among meteorological forcings attributes, NDVI, ET, and $R_{ECO}$ (Figure 3).
Both input data (e.g., meteorological forcings) and output data (ET and $R_{ECO}$) used for training and validation are
preprocessed for gap filling, smoothing, and data updating. HPM "learns" the complex space-time relationship among
meteorological forcings, NDVI, ET, and $R_{ECO}$ using a deep-learning-based module (deeply connected neural networks
and a long short-term memory recurrent neural network). HPM then can be used for ET and $R_{ECO}$ estimation at
sparsely monitored watersheds. Individual HPM models can be trained in two different ways using ET and $R_{ECO}$
information: with data obtained from flux towers ("data-driven HPM") or with outputs from 1-D physically-based
models ("mechanistic HPM"). In both cases, the models obtained with local data are then used to estimate ET and
$R_{ECO}$ at other sites in the same ecoregion (see Section 2.1). For ecoregions not represented by FLUXNET sites, it is
necessary to develop mechanistic HPM that enables ET and $R_{ECO}$ estimation over space and time.
HPM has several additional modules, including model development, model validation, and model prediction
modules. In the HPM model development module, deep-learning algorithms are trained with input features and
response data until a pre-defined "stopping criteria" (e.g., root mean squared error, RMSE) is met, indicating
subsequent training would lead to minimal improvement. In the validation module, estimation outputs from the
"trained HPM models" are compared with other ET and $R_{ECO}$ data obtained from other independent sites or
mechanistic models within the same ecoregion. Statistical measures, including adjusted $R^2$ and mean absolute error
(MAE), are computed to evaluate the performance of HPM models. In the predictive model module, meteorological
forcings data and remote sensing data are processed at target sites of interest, and the validated HPM model is used to
estimate ET and $R_{ECO}$ at these sites. ET and $R_{ECO}$ outputs estimated from HPM at sparsely monitored watersheds then
provide alternative datasets for process understanding within the target watersheds.



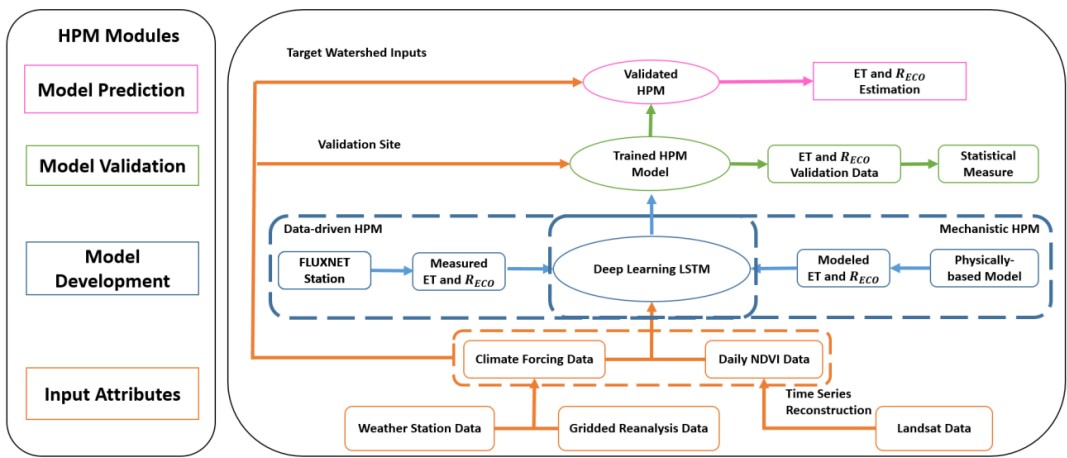

**Figure 3: Hybrid Predictive Model Framework. The HPM model mainly consists of four modules: Input Attributes, Model Development, Model Validation and Model Prediction, represented by rectangles with colors. Arrows represent the linkages among different modules. Choices of data-driven HPM or mechanistic HPM depend on the ecoregion of target watershed and data availability.**

Long short-term memory (LSTM, Hochreiter & Schmidhuber, 1997) is capable of identifying long-term dependencies between climate and environmental data. For example, Kratzert et al. (2018) successfully used LSTM to learn the long-term dependencies in hydrological data (e.g., storage effects within catchments, time lags between precipitation inputs and runoff generation) for rainfall-runoff modeling. LSTM has also been used for gap filling in hydrological monitoring networks in the spatiotemporal domain (Ren et al., 2019). In this study, the outputs (ET or $R_{ECO}$) denoted as $y$ are predicted from the input $x = [x_1, x_2, \ldots, x_T]$, consisting of the last $T$ consecutive time steps of attributes, such as meteorological forcings attributes (e.g., air temperature and precipitation) and remote sensing attributes (i.e., NDVI). In a recurrent neural network (RNN), $h_t$ represents the internal state at every time step $t$ that takes in current input value $x_t$ and previous internal state $h_{t-1}$, and is recomputed along the time axis using the following equation:

$$h_t = g(Wx_t + Uh_{t-1} + b), \tag{2}$$

where $g$ represents the hyperbolic tangent activation function, $W$ and $U$ are trainable weight metrices of the hidden state $h$, and $b$ is a bias vector. $W$, $U$ and $b$ are all trainable through optimization. LSTM introduces the cell state $c_t$, which makes LSTM powerful in identifying long-term dependencies in a statistical manner. The cell state $c_t$ has three gates structures, including "forget gates" (which determine what information from previous cell states will be forgotten), "input gates" (which determine what information will be conveyed from the forget gate) and "output gates" (which return information from cell state $c_t$ to a new state $h_t$). With these gate structures, the cell state $c_t$ controls what information will be forgotten, conveyed, and updated over time. The forget gate is formulated as follows:

$$f_t = \sigma(W_f x_t + U_f h_{t-1} + b_f), \tag{3}$$





where $f_t$ results in a value between 0 and 1 indicating the degree of information to be forgotten; $\sigma$ is the logistic
sigmoid function, $and$ $W_f, U_f$ and $b_f$ are trainable parameters. Next, the input gate decides which values will be
updated in the current cell state, and creates a vector of candidate values $\widetilde{c_t}$ in the range of (-1, 1) through a $tanh$ layer,
which will be used to update the current state. With the candidate values calculated from the current state, and the
information conveyed from the forget gate, we can calculate the current cell state as follows:

$$i_t = \sigma(W_i x_t + U_i h_{t-1} + b_i), \tag{4}$$

$$\widetilde{c_t} = tan\,h(W_{\tilde{c}} x_t + U_{\tilde{c}} h_{t-1} + b_{\hat{c}}), \tag{5}$$

$$c_t = f_t * c_{t-1} + i_t * \widetilde{c_t}, \tag{6}$$

where $i_t$ is the input gate that defines which information of $\widetilde{c_t}$ will be used to update the current cell state and is in the
range of (0, 1); $c_t$ represents the current cell state; and $W_{\tilde{c}}, U_{\tilde{c}}, b_{\tilde{c}}, W_i, U_i,$ $and$ $b_i$ are trainable parameters. Finally, the
output gate $o_t$ controls the information of cell state $c_t$ to a new hidden state $h_t$, which is computed using the following
equation:

$$o_t = \sigma(W_o x_t + U_o h_{t-1} + b_o), \tag{7}$$

$$h_t = \tanh(c_t) * o_t, \tag{8}$$

With the new hidden state calculated, ET and $R_{ECO}$ can be calculated using a one unit dense layer:

$$y_t = W_d h_t + b_d, \tag{9}$$

where $W_d$ and $b_d$ are additional trainable parameters. In summary, the LSTM unit calculates the internal state using
current meteorological forcings and remote sensing data at every time step. The forget gate, input gate, and output
gate decide what information from previous time steps will be kept, updated, and conveyed to the new hidden state.
Finally, with a single dense layer, the algorithm will output ET and $R_{ECO}$ estimation from the trained model.
A 70%-30% split between training and validation time series data was applied here, where the first 70% of the data
were used for HPM development as a learning process, and 30% of the data were used as validation sets at individual
sites. At the East River Watershed, HPM results were also validated with benchmark CLM outputs from Tran et al.
(2019) and FLUXNET measurements. We used the mean absolute error (MAE), and adjusted $R^2$ as the statistical
measure to determine model performance.

$$MAE = \frac{\sum_{i=1}^{n}|y_{predict} - y_{measured}|}{n}, \tag{10}$$

$$R^2 = 1 - \frac{SSE}{SS}, \tag{11}$$

where SSE represents the sum of squared errors, SS is the sum of squares of the response attributes (i.e., ET or $R_{ECO}$),
and n is the number of data points. In most models, the configuration of the neural networks includes a first LSTM
layer with 50 units, a second LSTM layer with 25 units, and a dense layer with 8 units having L2 regularizers and a
final output dense layer. Dropout layers are also embedded in the model to prevent overfitting. Other configurations





of networks may provide better estimation results; however, they are not assessed in this study. More information
about the LSTM-RNN method is provided by (Olah, 2015)
**3.2 Feature Selection**
Given data availability and the practicability of applying HPM to estimate ET and $R_{ECO}$ at sparsely monitored
watersheds, we also selected, constructed, and augmented certain attributes as features for HPM.
**3.2.1 Snow information**
In mountainous watersheds, snow dynamics significantly influence water and carbon fluxes. Because of the
difficulties in measuring snow time series over space, we did not directly use attributes such as snow water equivalent
as input to HPM. Instead, we separated precipitation data into snow precipitation (air temperature < 0) and rainfall
precipitation (air temperature > 0). This is in line with what has been used in hydrological models such as CLM
(Oleson et al., 2013). Note that for certain sites in this study, snow is not present (e.g., US-Ton). In order to capture
the dynamics of snow processes, such as accumulation and melting, we constructed a categorical variable (sn), as
follows:
$$sn = \begin{cases} 0, during\ snow\ accumulation;\ SWE > 0\ and\ SWE < peak\ SWE \\ 1, during\ snow\ melting; SWE > 0\ and\ SWE \leq peak\ SWE \\ 2, no\ snow; SWE = 0 \end{cases}, \quad (12)$$

Since data on peak SWE are rarely available because of the difficulties in measuring snow, we also define a
proxy categorical variable, $sn$. When no SWE measurements were available, we estimated $sn$ using air and soil
temperature data following Knowles et al. (2016), who found significant correlations between the day of peak snow
accumulation and first day of air temperature above 0 degrees Celsius, as follows:
$$sn = \begin{cases} 0, during\ snow\ accumulation;\ Air\ Temperature < 0 \\ 1, during\ snow\ melting; Air\ Temperature > 0\ while\ Soil\ Temperature \leq 0, \\ 2, no\ snow; Air\ Temperature\ and\ Soil\ Temperature > 0 \end{cases} \quad (13)$$

**3.2.2 Vegetation information**
To mitigate the long return periods of satellites and the presence of clouds, we reconstructed daily NDVI
values based on meteorological forcings data (e.g., air temperature, precipitation, radiation) using deep-learning
recurrent neural networks, leading to estimates of NDVI at daily temporal resolution. For example, Figure 4 represents
Landsat-derived NDVI and reconstructed NDVI values for two sites at the East River, CO watershed: Butte (ER-BT),
and Schofield Pass (ER-SP). Figure 4 reveals that based on meteorological forcings data only, the reconstructions
achieved an adjusted $R^2$ of 0.65. Though not ideal, as satellites continue to advance and more training data becomes
available, the accuracy of NDVI temporal reconstruction will increase.



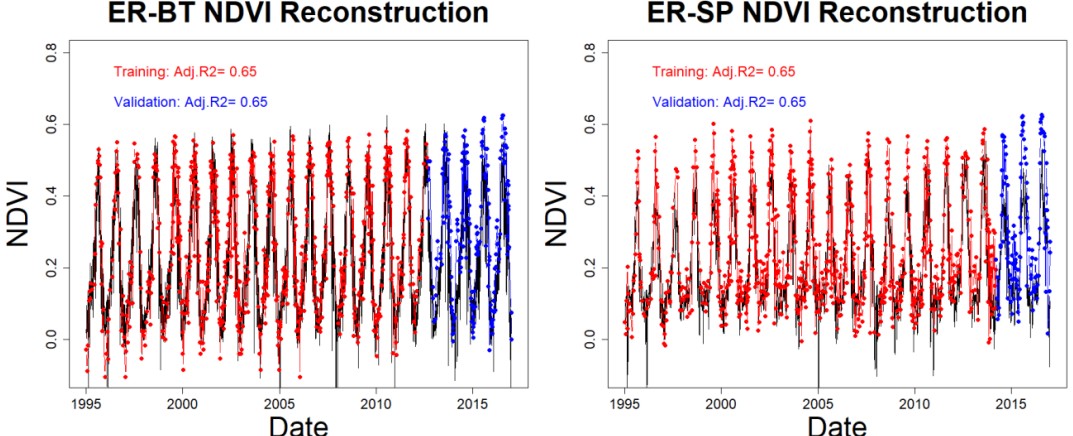


**Figure 4: Temporal reconstruction of NDVI at ER-BT (left) and ER-SP (right). Black line represents reconstructed daily**
**NDVI. Red points are used for training and blue points are used for validation**

**4. Results**

We tested HPM's capabilities using different use cases to explore different conditions. First, we tested the capability of HPM to estimate long-term temporal dependency among meteorological forcings, ET, and $R_{ECO}$ (presented in Section 4.1). Second, we validated HPM's capability to estimate the spatial distribution of ET and $R_{ECO}$ over space in selected watersheds, where we developed HPM using existing FLUXNET data (data-driven HPM, Section 4.2) or outputs from a mechanistic model (physical-model-based HPM, Section 4.3). Third, HPM was used to estimate ET and $R_{ECO}$ at selected sites within the East River Watershed and to distinguish how local factors (e.g., vegetation heterogeneity) influence ET and $R_{ECO}$ dynamics (Section 4.4). These four use cases illustrate and demonstrate how HPM can be developed and applied at target watersheds, where data are sparse.

**4.1 Use Case 1: ET and $R_{ECO}$ Time Series Estimation with HPM Developed at FLUXNET Sites**

Local HPMs were developed to estimate ET and $R_{ECO}$ using flux tower data obtained from FLUXNET sites listed in Table 1. Attributes used to train these individual HPM are documented in Table 3.

**Table 3. Attributes used for HPM development in Use Case 1**

| Site ID | Site Name | Attributes |
|---------|-----------|------------|
| US-NR1 | Niwot Ridge | Air Temperature, precipitation, net radiation, sn, NDVI, soil temperature |
| CA-Oas | Saskatchewan-Aspen | Air Temperature, precipitation, net radiation, sn, NDVI, soil temperature |
| CA-Obs | Saskatchewan-Black Spruce | Air Temperature, precipitation, net radiation, sn, NDVI, soil temperature |
| US-SRM | Santa Rita Mesquite | Air Temperature, precipitation, net radiation, NDVI, soil temperature |
| US-Ton | Tonzi Ranch | Air Temperature, precipitation, net radiation, NDVI, soil temperature |
| US-Var | Vaira Ranch-lone | Air Temperature, precipitation, net radiation, NDVI, soil temperature |
| US-Whs | Walnut Gulch Lucky Hills Shrub | Air Temperature, precipitation, net radiation, NDVI, soil temperature |
| US-Wkg | Walnut Gulch Kendall Grasslands | Air Temperature, precipitation, net radiation, NDVI, soil temperature |

The results, which are shown in Figure 5 and Table 4, reveal that the HPM approach was effective for estimating ET. Adjusted $R^2$ between the HTM estimates and flux tower measurements are above 0.85 for all sites,



and mean absolute errors are small at a level of $\sim 0.2\ mm/d$. Figure 5 displays the estimation of ET from HPM US-
NR1 and CA-OAS (other sites provided in supplementary material), and presents monthly mean ET values of
measurements, HPM estimations, and differences. The long-term trends in ET are well captured by HPM. At larger
temporal scales (monthly or yearly), HPM provides reasonable estimation of ET at these sites. However, short-term
fluctuations during the summer are also not well captured by ET, specifically at California sites during the periods
when plant transpiration and soil evaporation are constrained by soil moisture (Figure A2 panel a).

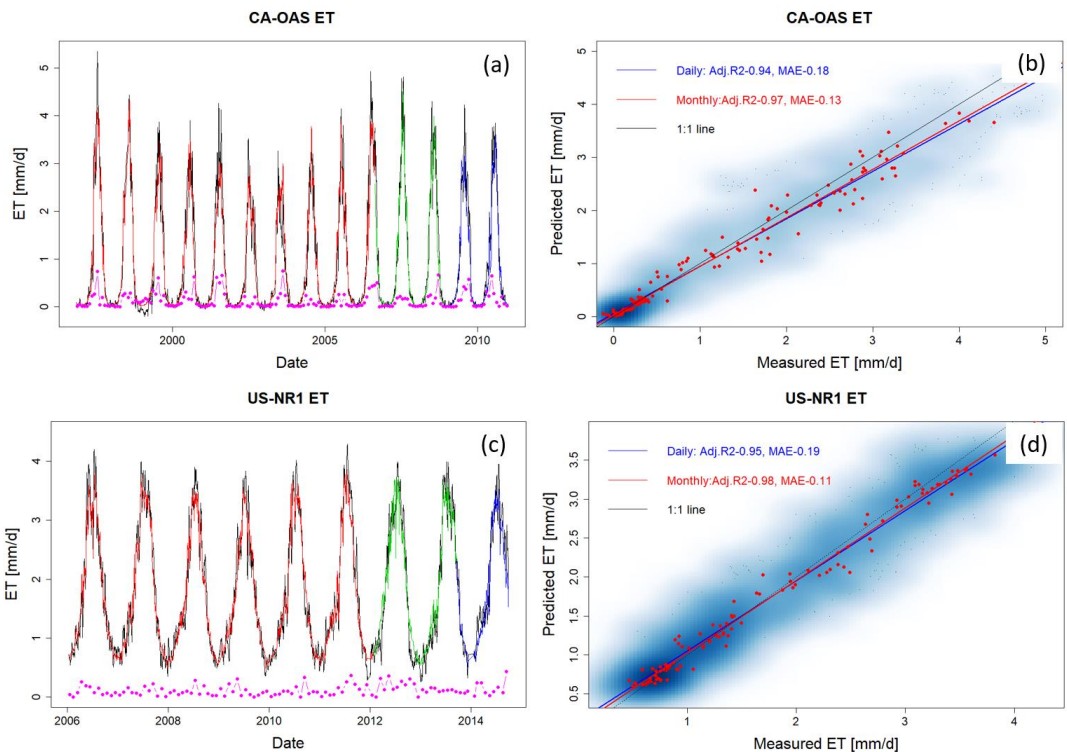


**Figure 5: ET estimation with data from FLUXNET sites at CA-OAS and US-NR1. Panels (a) and (c) illustrate the daily estimation of ET with red, green, and blue lines representing data used for training, validation, and prediction, respectively, and the black line showing the eddy covariance measurements. Pink points describe monthly mean difference between HPM estimation and measured data. Panels (b) and (d) show the scatter plots of daily (blue) and monthly (red) ET. Darker blue clouds represent greater density of data points. Results for other sites are included in supplementary materials below (Figures A1 and A2).**

Similarly, Table 4 and Figure 6 reveal that HPM was also effective in estimating $R_{ECO}$, leading to small *MAE*
and adjusted $R^2$ of 0.8 between estimated and measured $R_{ECO}$ except for US-Ton and US-Var. Figure 6 presents HPM-
estimated $R_{ECO}$ at US-NR1 and CA-OAS, with other sites presented in Figures A3 and A4. Long-term dynamics of
$R_{ECO}$ are also successfully captured by HPM; however, HPM underestimates $R_{ECO}$ during peak growing seasons. For
example, at US-NR1, error increased during the growing season, when estimates of $R_{ECO}$ are smaller than measured





$R_{ECO}$. While soil moisture can limit $R_{ECO}$ during peak growing season (Ng et al., 2014; Wang et al., 2014), the
developed HPM does not include soil moisture as a key attribute. As such, HPM underestimates $R_{ECO}$ during peak
growing season, leading to higher $MAE$ than other times of the year. In addition, HPM $R_{ECO}$ estimation at US-Ton
and US-Var show higher uncertainties (i.e., $MAE > 0.4$ and Adj. $R^2 < 0.8$), which also indicates that soil moisture
data is necessary to increase $R_{ECO}$ prediction accuracy in this ecoregion.

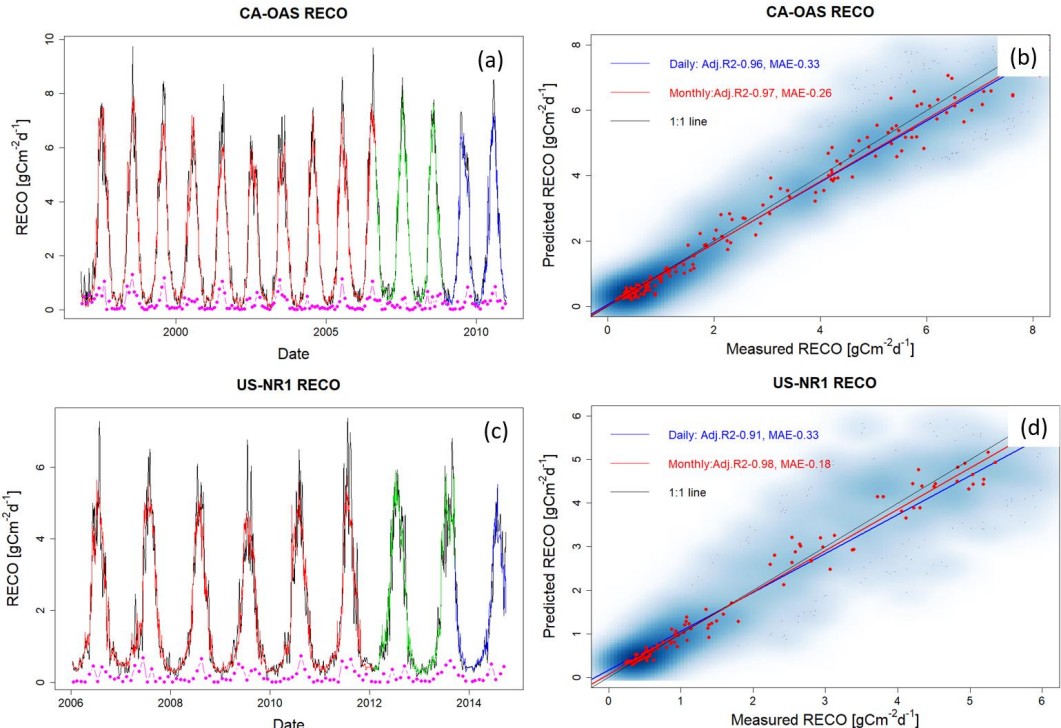


**Figure 6: $R_{ECO}$ estimation with data from FLUXNET sites at CA-OAS and US-NR1. Panels (a) and (c) present daily**
**estimation of $R_{ECO}$ with red, green, and blue lines representing data used for training, validation, and prediction, and the**
**black line shows the eddy covariance measurements. Pink points describe monthly mean difference between HPM**
**estimation and measured data. Panels (b) and (d) show the scatter plots of daily (blue) and monthly (red) $R_{ECO}$. Darker**
**blue clouds represent greater density of data points. Results for other sites are included in supplementary materials below**
**(Figures A3 and A4).**

**Table 4: Statistical measures of HPM estimation of ET and $R_{ECO}$**

| Site ID | Train MAE -ET [$mm/d$] | Test MAE - ET [$mm/d$] | Train Adj. $R^2$ - ET | Test Adj. $R^2$ - ET | Train MAE $-R_{ECO}$ [$gCm^{-2}d^{-1}$] | Test MAE $-R_{ECO}$ [$gCm^{-2}d^{-1}$] | Train Adj. $R^2$ $-R_{ECO}$ | Test Adj. $R^2$ $-R_{ECO}$ |
|---|---|---|---|---|---|---|---|---|
| US-NR1 | 0.19 | 0.11 | 0.95 | 0.98 | 0.33 | 0.18 | 0.91 | 0.98 |
| CA-Oas | 0.18 | 0.13 | 0.94 | 0.97 | 0.33 | 0.26 | 0.96 | 0.97 |
| CA-Obs | 0.12 | 0.09 | 0.95 | 0.96 | 0.29 | 0.25 | 0.96 | 0.97 |
| US-SRM | 0.22 | 0.17 | 0.92 | 0.94 | 0.24 | 0.19 | 0.80 | 0.87 |
| US-Ton | 0.22 | 0.17 | 0.92 | 0.94 | 0.43 | 0.36 | 0.76 | 0.82 |





| US-Var | 0.15 | 0.12 | 0.92 | 0.95 | 0.49 | 0.38 | 0.81 | 0.88 |
| US-Whs | 0.13 | 0.09 | 0.93 | 0.96 | 0.12 | 0.09 | 0.84 | 0.89 |
| US-Wkg | 0.19 | 0.15 | 0.87 | 0.91 | 0.18 | 0.15 | 0.85 | 0.91 |


**4.2 Use Case 2: Ecoregion-Based, Data-Driven HPM Model for ET and $R_{ECO}$ Estimation**

While the effort and cost involved in establishing flux towers naturally limit the spatial coverage of obtained
measurements, point scale measurements from one FLUXNET station provides representative information about
ecosystem dynamics at other locations within the same ecoregion. In this section, we explored the use of a data-driven
HPM trained with one FLUXNET station to estimate ET and $R_{ECO}$ at other locations within the same ecoregion. To
test this approach, we first trained HPM at a selected FLUXNET stations and validated these HPM models at other
FLUXNET stations (ET and $R_{ECO}$ data at testing sites were only used for comparison with HPM prediction) within
the same ecoregion. Specifically, we developed HPM models at US-Ton, CA-Oas and US-Wkg, and provided ET and
$R_{ECO}$ estimations at US-Var, CA-Obs and US-Whs at three ecoregions, respectively.
Table 5 summarizes how we developed the data-driven HPM models for spatially distributed estimation of
ET and $R_{ECO}$ as well as the corresponding statistical summaries.  The estimation led to an adjusted $R^2$ greater than
0.85 for US-Obs and US-Whs and 0.70 for US-Var. Figures 7 and 8 present the time series of HPM-estimated ET and
$R_{ECO}$ compared to measurements from flux towers. The figures show that HPM captures the seasonal and longer-term
dynamics of ET and $R_{ECO}$ well, as indicated by the high adjusted $R^2$. However, we observed an increased error in
HPM-based estimations compared to measurements during peak growing seasons (e.g., a 0.5 mm discrepancy in June
mean ET). Higher prediction accuracy for the two ecoregions presented by US-Whs and CA-Obs are observed
compared to US-Ton, which indicates other attributes (e.g., soil moisture) are necessary to improve prediction
accuracy, especially for sites limited by moisture conditions. Although the prediction accuracy is not as high as Use
Case 1 (Section 4.1), this use case demonstrates that HPM can learn the complicated relationships between responses
and features successfully, and that a local data-driven HPM can be used to fuse with data from other subsites for long-
term estimation of ET and $R_{ECO}$ within the same ecoregions.



**Figure 7. ET estimation at CA-Oas (a), US-Var (c), and US-Whs (e) with HPM trained at US-Ton, US-Wkg, and CA-Oas, respectively. Red and black lines represent HPM estimation and real measurements, with green points denoting the monthly mean difference between HPM estimationss and measurements. Panels (b), (d), and (f) show the scatter plots of daily (blue) and monthly (red) ET at these three sites. Darker blue clouds represent greater density of data points.**



**Figure 8.** $R_{ECO}$ estimation at CA-Oas (a), US-Var (c), and US-Whs (e) with HPM trained at US-Ton, US-Wkg, and CA-Oas, respectively. Red and black lines represent HPM estimations and real measurements; green points denote the monthly mean difference between HPM estimation and measurements. Panels (b), (d), and (f) show the scatter plots of daily (blue) and monthly (red) $R_{ECO}$ at these three sites. Darker blue clouds represent greater density of data points.

**4.3 Use Case 3: Ecoregion-Based, Mechanistic HPM Estimation of ET**

Mechanistic HPM, which is trained with ET estimates from 1-D physically-based-model simulations, provides an avenue for estimating ET in ecoregions where direct measurements from eddy covariance tower are not





available. In order to test the effectiveness of the mechanistic HPM, we focused on the three SNOTEL stations and
US-NR1, which locates in the "Western Cordillera" ecoregion. Mechanistic HPM is coupled with CLM simulations
at these sites (Tran et al., 2019). To ensure the CLM physically-based-model simulations can provide alternative
datasets to develop mechanistic HPMs, we compared CLM estimation and direct measurements of ET at US-NR1
(Figure S2). The consistent results between measured ET and CLM-estimated ET (adjusted $R^2 = 0.88$; $k = 0.95$)
indicate independent CLM simulations can be effectively used to develop the mechanistic HPM.
We applied mechanistic HPM trained with 1-D CLM developed at ER-BT (Tran et al., 2019) to estimate ET
at sites classified as part of the same ecoregion (i.e., ER-SP, ER-PK and US-NR1). We then compared ET estimation
from HPM to independent CLM-based ET estimations at ER-SP and ER-PK and to direct measurements at US-NR1.
Figure 9 shows a high consistency between HPM estimation and the validation data. For all scenarios, an adjusted $R^2$
of 0.8 or greater is observed (Table 5), which strongly indicates that mechanistic HPM can provide accurate ET
estimation at sites of similar ecoregions. These results suggest the broad applicability of mechanistic HPM to estimate
ET based on ecoregion characteristics. This approach is expected to be particularly useful for regions where flux
towers are difficult to install or where measured fluxes are not representative of the landscape, such as in mountainous
watersheds.
**Table 5. Statistical summary of HPM estimation over space with FLUXNET sites and SNOTEL stations with CLM**

| Target Site | Training Site | Level II Ecoregion | ET MSE (monthly)$[mm/d]$ | ET Adj. $R^2$ | $R_{ECO}$ MSE(monthly)$[gCm^{-2}d^{-1}]$ | $R_{ECO}$ Adj. $R^2$ |
|---|---|---|---|---|---|---|
| CA-Obs | CA-Oas | Boreal Plain | 0.39 | 0.88 | 0.36 | 0.97 |
| US-Var | US-Ton | Mediterrean California | 0.34 | 0.70 | 0.67 | 0.70 |
| US-Whs | US-Wkg | Western Serra Madre Pidemont | 0.13 | 0.94 | 0.17 | 0.85 |
| ER-SP | ER-BT | Western Cordillera | 0.20 | 0.92 | - | - |
| ER-PK | ER-BT | Western Cordillera | 0.24 | 0.90 | - | - |
| US-NR1 | ER-BT | Western Cordillera | 0.23 | 0.90 | | |





**Figure 9. HPMs trained with CLM simulation at ER-BT are used to estimate ET at ER-SP, ER-PK, and US-NR1. Panels (a), (c), and (e) display HPM estimation of ET (red lines), as well as independent CLM estimation at ER-SP, ER-PK, and eddy covariance measurements at US-NR1 (black lines). Panels (b), (d), and (f) show the scatter plots of daily (blue) and monthly (red) ET at these three sites. Darker blue clouds represent greater density of data points.**

**4.4 Exploration of How ET and $R_{ECO}$ Varies with Meteorological forcings and Vegetation Heterogeneity at the East River Watershed**

ET and $R_{ECO}$ estimated from the HPM model at the mountainous East River Watershed in CO enabled us to analyze how vegetation heterogeneity and meteorological forcings heterogeneity influence estimated ET and $R_{ECO}$





dynamics, and to identify limitations in the developed approach for estimating ET and $R_{ECO}$ across mountainous and
heterogeneous watersheds.

NDVI time-series data provide high-resolution (30m) information about vegetation variability across the East

River Watershed. The spatial distribution of vegetation cover presented in Figure 2 (from Falco et al. 2019) enables
us to distinguish different patches of deciduous forests, evergreen forests, meadow grassland and riparian shrublands
and retrieve corresponding NDVI time-series.  Figure 10 shows Landsat-derived and reconstructed NDVI values for
the four different vegetation types within the East River Watershed. Evergreen forests have an extended growing
season compared to deciduous forests. However, peak NDVI is smaller in evergreen forests compared to deciduous
forests. NDVI ranges from 0.2 to 0.6 for evergreen forests, whereas larger fluctuations in NDVI are observed for
deciduous forests (-0.2 to 0.8).  The NDVI values during the winter are likely sensing both snow and forest density,
due to pixel spatial averaging from Landsat images. Similar to Qiao et al. (2016), we also found that the NDVI of
deciduous forests exhibits a significant increase during the growing season, followed by a sharp decline (likely caused
by defoliation), and that evergreen forests had a more stable NDVI.

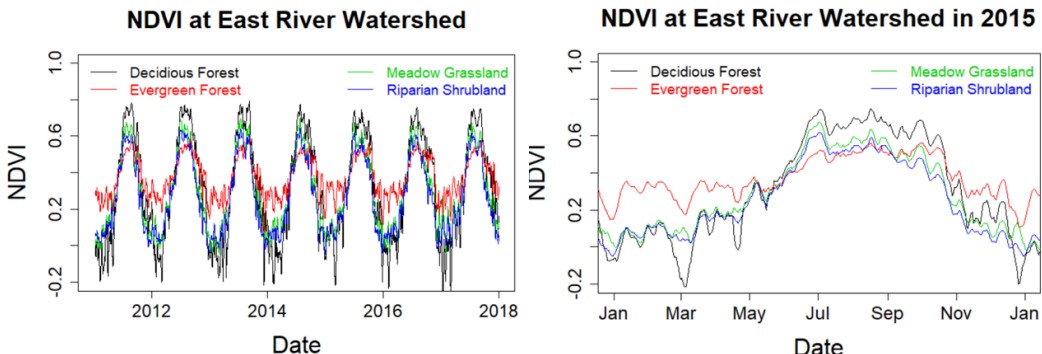


**Figure 10: Reconstructed NDVI time series at selected locations in the East River Watershed for 2011 to 2018 (panel a) and**

**for 2015 (panel b, normal water year). Black, red, green, and blue lines represent the time series of NDVI for deciduous**

**forests, meadow grasslands, evergreen forests and riparian shrubland, respectively**.

HPM-estimated ET and $R_{ECO}$ also show different dynamics in evergreen forests and deciduous forests. Figure

11a and 11b present the time series of estimated ET and $R_{ECO}$ associated with deciduous forests, respectively. Figure
11c and d present the ET and $R_{ECO}$ differences between deciduous forests sites and sites with other vegetation (e.g.,
evergreen forests shown in red). Before peak growing season, the ET of evergreen forests is about 10% greater than
deciduous forests, whereas ET of deciduous forests during peak growing season is greater than evergreen forests.
After growing season, the NDVI of deciduous forests is less than 0.2 (loss of leaves) compared to the NDVI of
evergreen forests. Before peak growing season, $R_{ECO}$ of evergreen forests is slightly greater than deciduous forests.
During peak growing season, $R_{ECO}$ of deciduous forests is around 17% greater than $R_{ECO}$ of evergreen forests. Total
annual ET between evergreen and deciduous forests is very close (DF1: 535 to 573 mm and EF1: 532 to 569 mm
across 7 years in this study). Total annual $R_{ECO}$ of evergreen forests is smaller than deciduous forests (DF1: 642 to





698 $gCm^{-2}d^{-1}$ and EF1: 592 to $639 gCm^{-2}$). The ET estimation at East River Watershed is comparable to Mu et al.
(2013), where ET is computed based upon the logic of the Penman-Monteith equation and MODIS remote sensing
data. At the East River Watershed, data retrieved from Mu et al. (2013) indicate annual ET ranges from 554 to 585
mm at deciduous forests sites and 540 to 593 mm at evergreen forests sites. The $R^2$ between 8-day aggregated HPM-
based ET estimation and data retrieved from Mu et al. (2013) achieves 0.65 (Figure S1). Berryman et al. (2018)
developed a random forest model to predict growing season soil respiration at subalpine forests in the Southern Rocky
Mountain ecoregions. Their results suggest a consistent respiration rate from 2004 to 2006, with 150-day sums of
$542.8, 544.3 \ and \ 536.5 \ gCm^{-2}$, respectively, with a mean measured growing season respiration across sites and
years of 3.37 $gCm^{-2}$. HPM-based $R_{ECO}$ estimation is also comparable to what Berryman et al. (2018) discovered,
with growing season $R_{ECO}$ ranging between 555 to 607 $gCm^{-2}$ and mean growing season $R_{ECO}$ ranging between
$3.01 \ to \ 3.30 \ gCm^{-2}$. While we currently do not have a time-series measurement of ET and $R_{ECO}$ at the East River
Watershed for validation, our results are comparable to other studies that focus on sites within the same ecoregion
(e.g., Berryman et al., 2018).



**Figure 11: ET (a) and $R_{ECO}$ (b) estimation for the deciduous forest site DF1 at the East River Watershed. Panels (c) and (d) show the differences in ET and $R_{ECO}$ among various vegetation types and DF1. Red, green, and blue lines represent the differences in evergreen forest, meadow, and riparian shrubland compared to DF1. Panels (e) and (f) zoom into 2015 to better display seasonal variations.**

ET and $R_{ECO}$ estimation at the East River Watershed from the HPM model further enabled us to assess the impacts of small-scale (e.g., hillslope scale) heterogeneity in vegetation type on ET and $R_{ECO}$ dynamics. Figure 12





shows the absolute value of monthly mean difference in ET (Fig. 12a and Fig. 12b) and $R_{ECO}$ (Fig. 12c and Fig. 12d)
across SNOTEL stations (ER-BT, ER-SP and ER-PK) and within selected East River locations. A comparison of
meteorological forcings data within selected East River locations and across SNOTEL stations are given in Figure S3.
We observed 2.5 times greater differences in ET across SNOTEL stations compared to the sites within the East River
watershed, whereas the differences in $R_{ECO}$ across SNOTEL stations are at the same level compared to the sites within
East River Watershed (around 0.8 $gCm^{-2}$). This result indicates small-scale meteorological forcings and vegetation
heterogeneity are the major controls of differences in ET and $R_{ECO}$ at the East River Watershed.

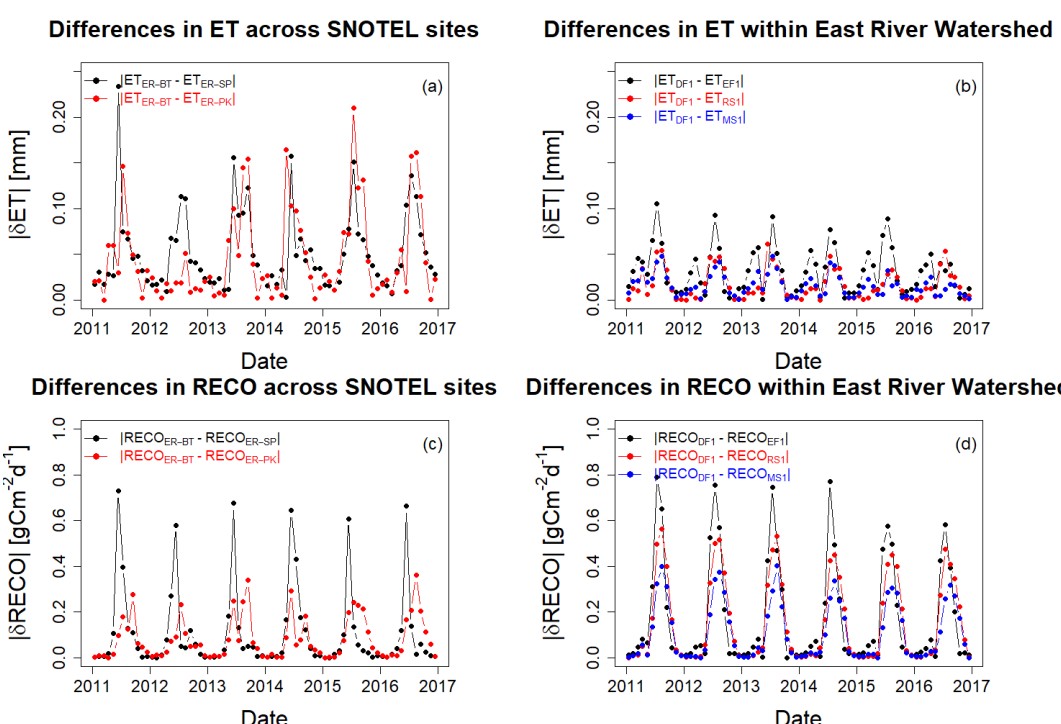

**Figure 12. Absolute differences in monthly mean ET and $R_{ECO}$ across SNOTEL stations and within East River Watershed.**
**Panels (a) and (c) describe the absolute differences in monthly mean ET and $R_{ECO}$ between ER-BT, ER-SP, and ER-PK.**
**Panels (b) and (d) describe the absolute differences in monthly mean ET and $R_{ECO}$ within East River Watershed between**
**deciduous forest (DF1), evergreen forest (EF1), meadow (MS1), and riparian shrubland (RS1).**
**5. Discussion**

Our study demonstrates that HPM provides reliable estimations of ET and $R_{ECO}$ under various climate and

vegetation conditions, including data-based HPMs that are trained with FLUXNET data and physical-model-based
HPMs that are coupled with simulations results from mechanistic models (i.e., CLM in our case). With 70% of the
data used for training (model development), ET and $R_{ECO}$ estimation from HPM achieves an adjusted $R^2$ of 0.9
compared to eddy covariance measurements. With this high estimation accuracy, we demonstrated that this approach



could be used for predicting ET and $R_{ECO}$ over time. HPM is capable of "learning" the complex interactions among
meteorological forcings, vegetation dynamics, and water and carbon fluxes. The underlying relationships acquired by
HPM can serve as a local ecohydrological model for long-term monitoring of ET and $R_{ECO}$, with the aid of remote
sensing data, and can fill in gap data during occasional equipment failure.

HPM was also successful at estimating the spatial distribution of ET and $R_{ECO}$ through exploiting an

ecoregion concept. Using the representative FLUXNET sites in different ecoregions, HPM provided estimates of ET
and $R_{ECO}$ at locations using learned relationships from other sites having the same ecoregion classification. For
conditions where no FLUXNET sites are within the same ecoregion, our study showed that physically-based models
that can utilize weather forcings data can provide alternatives for developing mechanistic HPM to estimate ET and
$R_{ECO}$. We found that HPM performance was more reliable when trained and applied at different watersheds in the
same ecoregion. For example, HPM that only relies on energy-related parameters was able to successfully estimate
ET and $R_{ECO}$ at US-NR1 and CA-OAS, where radiation and temperature are key components that regulate ET and
$R_{ECO}$ dynamics. However, HPM with the same input features do not yield desired results at sites limited by water
conditions (e.g., US-Ton and US-Var), due to lack of soil moisture data. This change indicates that parameter
optimization and attributes selection may be needed for sites that are limited by moisture conditions, because important
features can be subject to local conditions that potentially lower HPM performance.

We confirmed the important role of small-scale vegetation heterogeneity in modeling ET and $R_{ECO}$ dynamics,

which further enabled us to better understand ecosystem dynamics at the East River Watershed. As indicated from
NDVI time series (Fig 10), evergreen forests have a longer growing season compared to deciduous forests; however,
deciduous forests have greater peak NDVI values. Correspondingly, we also observed an earlier increase in ET and
$R_{ECO}$ for evergreen forests (before May), but larger ET and $R_{ECO}$ for deciduous forests during peak growing season
(around June and July). Annual ET between deciduous forests and evergreen forests are not statistically different,
which is similar to (Berryman et al., 2018; Mu et al., 2013). Annual $R_{ECO}$ differences between evergreen forests and
deciduous forests are around 50 $gCm^{-2}$, which is comparable to Berryman et al. 2018). Similar dynamics were also
observed at regions that are have different climate conditions. Through assessing the differential mechanisms of
deciduous forests and evergreen forests at various sites under Mediterranean climates, Baldocchi et al. (2010) found
that deciduous forests had a shorter growing season, but showed a greater capacity for assimilating carbon during the
growing season. Evergreen forests, on the other hand, had an extended growing season but with a smaller capacity for
gaining carbon. These results were identified through analyzing the relationships among leaf ages, leaf nitrogen level,
leaf area, and water use efficiencies of these tree species at the selected Mediterranean sites. Older leaves tend to have
smaller leaf nitrogen and stomata conductance, and thus evergreen forest ET and $R_{ECO}$ are smaller during the peak
growing season compared to deciduous forests, yet maintain a relatively high level before the peak growing season or
during defoliation. Hu et al. (2010) analyzed flux data at US-NR1 to determine the relationships between growing
season lengths and carbon sequestration, and found that extended growing season length resulted in less annual $CO_2$
uptake. They also found that the duration of growing seasons substantially decreases snow water storage, which
significantly decreases forest carbon uptake. While we were not able in this study to assess the differential advatanges





and physiological mechanisms among vegetation types, HPM-based estimation of ET and $R_{ECO}$ presented similar
dynamic trends to those found in Berryman et al. (2018); Hu et al. (2018); and Mu et al. (2013).

Microclimate and small-scale heterogeneity in meteorological forcings attributes control the magnitude and

timing of ET and $R_{ECO}$ dynamics. For example, other field observations along the Rocky Mountain ranges have shown
that south-facing hillslopes have significantly earlier snowmelt compared to north-facing hillslopes (Kampf et al.,
2015; Webb et al., 2018), which are hypothesized to result in significant differences in ET and $R_{ECO}$ dynamics. As a
result, estimation of small-scale ET and $R_{ECO}$ dynamics requires high spatial resolution meteorological inputs, which
is currently a challenge. We originally intended to investigate aspect impacts on ET and $R_{ECO}$ dynamics at East River
Watershed by selecting East River sites with different slope orientations. However, small-scale meteorological-
forcings heterogeneity and microclimate were not available due to the relatively low spatial resolution of
meteorological forcings inputs (DAYMET, 1 km scale). While DAYMET data suggest that differences in air
temperature and solar radiation are very small for sites located at different portions of the watershed, the three weather
stations at the site reveal that spatial heterogeneity in meteorological forcing attributes do exist, especially air
temperature (Figure S4). Even though the small-scale meteorological forcings heterogeneity is partly embedded in
NDVI time series, the heterogeneity in ET and $R_{ECO}$ estimated from HPM at the East River Watershed is potentially
underestimated, due to the insufficient spatial resolution of meteorological inputs. In addition to limitations imposed
from the spatial resolution, uncertainties in meteorological inputs can also result in large errors (i.e., >20% MAE) and
reduce accuracy by 10-30% in ET and $R_{ECO}$ estimations as suggested by Mu et al. (2013) and Zhang et al. (2019).
Thus, there is still a significant need for high-spatial-resolution meteorological-forcing data products, such as data
provided by the Surface Atmosphere Integrated Field Laboratory (SAIL) that can capture small-scale heterogeneity
for implementing into HPM, which will then enable us to better assess the governing factors that regulate small-scale
heterogeneity in ET and $R_{ECO}$.

In addition to the quality of meteorological data, HPM is also influenced by remote sensing inputs accuracy.

Incorrectly calculated or pixel-averaged NDVI values from Landsat images can greatly alter HPM outputs for ET and
$R_{ECO}$. Satellite images with different cloud cover have a slight influence over the NDVI values calculated, which do
not represent real-time vegetation conditions. Algorithms used to reconstruct daily NDVI time series are also subject
to uncertainties. However, with recent advances in remote sensing and satellite technologies (McCabe et al., 2017),
the spatial and temporal resolution should greatly increase in the future (i.e., 3 m resolution and daily). These advances
will lead to more accurate classification of vegetation types and NDVI calculations, which are expected to decrease
uncertainty associated with flux estimation

Another source of uncertainty in HPM arises from the choice of hybrid approaches and any parameter

uncertainties in mechanistic models. Since HPM relies on accurate ET and $R_{ECO}$ inputs from flux towers or
mechanistic models, any uncertainties in measuring or modeling ET and $R_{ECO}$ will propagate to HPM. If HPM is
developed with a mechanistic model that has such missing components, these biases will be passed on to HPM
estimation of ET and $R_{ECO}$. Parameter and conceptual model uncertainties in mechanistic models also restrict HPM's
ability to "learn" the ecosystem dynamics. In order to reduce potential biasedness, we trained data-based HPM and



physical-model-based HPM upon long time series (e.g., > 5 years) with quality assessed data or simulation results,
which also enables HPM to better memorize long time dependencies of ecosystem dynamics. Though the
quantification of uncertainties remains challenging, efforts have been made to lower these uncertainties using the
technical advances described here.
**6. Conclusion**
In this study, we developed and tested a Hybrid Predictive Modeling (HPM) approach for ET and $R_{ECO}$
estimation, with a focus on mountainous watersheds. We developed individual HPM models at various FLUXNET
sites and at sites where data can supports the proper development of a mechanistic model (e.g., CLM). These models
were validated against eddy covariance measurements and CLM outputs. We further used these models for ET and
$R_{ECO}$ estimation at watersheds within the same ecoregion to test HPM's capability of providing estimation over space,
where only meteorological forcings data and remote sensing data were available. Lastly, we applied the HPM to
provide long-term estimation of ET and $R_{ECO}$ and test the sensitivity of HPM to various vegetation types at various
sites within the East River Watershed.
Given the promising results of HPM, this work offers an avenue for estimating ET and $R_{ECO}$ using easy-to-
acquire or commonly available datasets. This study also suggests that the spatial heterogeneity of meteorological
forcings and vegetation dynamics have significant impacts on ET and $R_{ECO}$ dynamics, which may be currently
underestimated due to typically coarse spatial resolution of data inputs. Parameters related to energy and soil moisture
conditions can be implemented into HPM to increase HPM's accuracy, especially for sites limited by soil moisture
conditions. Lastly, it should be pointed out that HPM is not restricted to estimation of ET and $R_{ECO}$ only. We focused
here on developing HPM for ET and $R_{ECO}$, but HPM also has great potential for estimating other parameters important
for water and carbon cycles. Indeed, other attributes, such as GPP and sensible heat flux, might also be accurately
captured and represented with HPM, given the right choice of features.
**Data availability.** The data used in this study are from publicly available datasets. FLUXNET measurements can be
accessed at https://FLUXNET.fluxdata.org. SNOTEL data are available at https://www.wcc.nrcs.usda.gov/snow/.
DAYMET data can be found at (Thornton et al., 2017) or via Google Earth Engine. Landsat data are available on
Google Earth Engine. All data and simulated results associated with this article can be found at https://data.ess-
dive.lbl.gov/view/doi:10.15485/1633810.
**Acknowledgement.** This material is based upon work supported as part of the Watershed Function Scientific Focus
Area funded by the U.S. Department of Energy, Office of Science, Office of Biological and Environmental Research
under Award Number DE-AC02-05CH11231. We thank Haruko Wainwright and Bhavna Arora for providing
comments on East River estimations. We also greatly appreciate all the guidance provided by Professor Yoram Rubin
and Professor Dennis Baldocchi at UC Berkeley to the first author. We also acknowledge the Jane Lewis Fellowship
Committee of the UC Berkeley for providing fellowship support to the first author.

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

**CA-OBS ET**

**CA-OBS ET**

**US-WKG ET**

**US-WKG ET**

**US-SRM ET**

**US-SRM ET**

Figure A1: ET estimation with data from selected FLUXNET sites at CA-OBS, US-Wkg, and US-SRM. Panels (a), (c), and (e) present daily estimations of ET with red, green, and blue lines representing data used for training, validation, and prediction, respectively, and the black line representing the eddy covariance measurement. Pink points describe monthly mean difference between HPM estimation and measured data. Panels (b), (d), and (f) show the scatter plots of daily (blue) and monthly (red) ET. Darker blue clouds represent greater density of data points.



**Figure A2: ET estimation with data from selected FLUXNET sites at US-Ton, US-Var, and US-Whs. Panels (a), (c), and (e) present daily estimations of ET with red, green, and blue lines representing data used for training, validation, and prediction, respectively, and the black line representing the eddy covariance measurement. Pink points describe monthly mean difference between HPM estimation and measured data. Panels (b), (d), and (f) show the scatter plots of daily (blue) and monthly (red) ET. Darker blue clouds represent greater density of data points.**

**Figure A3:** $R_{ECO}$ **estimation with data from selected FLUXNET sites at CA-OBS, US-Wkg, and US-SRM. Panels (a), (c),**
**and (e) present daily estimations of** $R_{ECO}$ **with red, green, and blue lines representing data used for training, validation, and**
**prediction, respectively, and the black line is eddy covariance measurement. Pink points describe the monthly mean**
**difference between HPM estimation and measured data. Panels (b), (d), and (f) show the scatter plots of daily (blue) and**
**monthly (red)** $R_{ECO}$. **Darker blue clouds represent greater density of data points.**





**Figure A4:** $R_{ECO}$ estimation with data from selected FLUXNET sites at US-Ton, US-Var, and US-Whs. Panels (a), (c), and (e) present daily estimations of $R_{ECO}$ with red, green, and blue lines representing data used for training, validation, and prediction, respectively, and the black line representing the eddy covariance measurement. Pink points describe monthly mean difference between HPM estimation and measured data. Panels (b), (d), and (f) show the scatter plots of daily (blue) and monthly (red) $R_{ECO}$. Darker blue clouds represent greater density of data points.