# Peer review of "A Deep-Learning Hybrid-Predictive-Modeling Approach for Estimating Evapotranspiration and Ecosystem Respiration"

_Hydrology and Earth System Sciences, 2020_

## Referee Comment (RC1) · Anonymous Referee #1 · 14 Aug 2020

This study tested and validated a hybrid predictive modeling (HPM) approach at eight flux tower sites and three snow measurement sites in western North America. Modeled predications of annual evapotranspiration and ecosystem respiration fluxes were significantly correlated with observations but less accurate at sub-annual resolution. These results are very promising but also demonstrate some limitations of the current HPM approach.

General comments: This is a thorough analysis and very promising study for the application of hybrid models to simulate ecosystem fluxes. It's also a well written manuscript. However, minimal effort has been devoted to generating the type of

process-based/transferrable information that's expected from a top journal. Instead, the discussion is generally couched in terms of supporting previous work to emphasize model performance. In my opinion, some re-interpretation of results is needed to move this beyond a basic model validation.

A good recent example is Wieder et al. 2017 that also compared CLM (point mode) to flux tower observations in complex terrain. Although their study only considered one site, they explicitly focused on periods of relative agreement versus disagreement between the modeled and observed fluxes to yield broadly testable ecohydrological hypotheses. Given the multi-site focus of the current study, I don't think the same level of detailed inquiry is required, but additional synthetic analysis would increase the scope and the subsequent impact of this work.

Relatedly, Figures 4-9 all show similar long-term timeseries data with scatterplots that lend themselves to similar interpretations in terms of R2 of MAE. These are useful, but perhaps they could be condensed and/or supplemented with other figure types that were more conducive to process-based interpretation. For example, I found Figures 11e and 11f fascinating insofar as they highlighted seasonal differences between vegetation types, but little explanation was provided to "unpack" these results (grasslands and shrublands not even mentioned). Likewise, Figures 12a and 12b present a rich opportunity to speak to differences between the biophysical controls on ET at the SNOTEL and East River sites. Some of the specific factors I'm left wondering about are differences in snow accumulation and melt between sites, evaporation versus transpiration, and heterotrophic versus autotrophic respiration. I understand that you don't have all these measurements, but you've generated a lot of suggestive data that could be leveraged to push this field of research.

Specific comments: L66-67: Reading back through the manuscript, this seems at odds with the practice of using single flux towers to represent the larger ecoregion (section 4.2). I don't actually have a problem with that research design, but this heterogeneity descussion may not be the best way to set things up.

[Figure]

L73-75: Also uneven hydrologic distribution due to lateral flow in complex terrain (e.g., Chang et al. 2018) that results in heterogeneous fluxes.

L109: Has NDVI been defined?

L142-144; L365-367: After reading through the manuscript once, I'm not convinced this objective was met or even really addressed, which was confusing because I kept expecting to come across these results. The small-scale heterogeneity results must be expanded or else it may not be a fatal flaw to just remove this language/objective if the analysis didn't work out (as you intimate on L574-577). In any case, the current manuscript introduction/objectives/results are inconsistent with respect to the degree of focus on this topic.

L143: Replace "CO" with "Colorado, USA" for the global audience.

L150: I'm curious how you defined "mountainous watersheds" for this study. I've been to the Walnut Gulch sites and they didn't strike me as the least bit mountainous. Also, with respect to my comment on L142-144, how important is the "mountainous" aspect anyway? I understand the broader impacts for water resources, but you'd reach a wider audience if the results were presented in a more general way. I see advantages and disadvantages to both mountain-specific and general analyses, but details/justification (mountain) or else re-framing (general) is needed in either case.

L162: How were the eight FLUXNET stations selected? Some justification needed here. Was it to facilitate the paired approach in section 4.2?

L164: Table 1 indicates that the Saskatchewan sites are colder than US-NR1.

Table 1: I assume the periods of record are truncated at 2015 because you used the FLUXNET2015 product? This should be specified. Watch significant figures throughout this table.

L227: Why was it necessary to treat this site different than the others? Please provide details about this "cleaning" procedure and why it was needed.

L367: The previous text makes it sound like three (not four) cases – confusing.

Table 3: You probably don't a need a table just to say that "sn" was included at three of the eight sites. Especially because you already have so many display items.

L378-380: I'm very curious as to whether this was also the case at the seasonally dry Walnut Gulch sites? If so, it speaks to systematic bias where the model captures ET dynamics during energy-limited but not water-limited periods. This strikes me as a major result (see general comments) and could be leveraged to make recommendations about the input variables that are necessary for various systems.

L393-395: Wouldn't the model overestimate (not underestimate) Reco if it can't account for moisture limitation during this time? Please clarify.

L408-411: Seemingly contradicts L66-67.

L493: Units mismatch.

L495-505: Discussion.

L516-518: In my mind, this is a missed opportunity to gain process-based (and thus transferrable) insight. What about these sites could factor into ET differences that are so much greater than the Reco differences? See general comments.

L544-546: How hard would it be to add moisture into the model? Why wasn't it added in the first place? I'm not suggesting that you re-do the analysis, but readers will be very interested in this information.

L563-568: It's not clear to me what model results "present similar dynamic trends" to the moisture limitation invoked by Hu et al. 2010 (and a host of larger scale, more recent work). My current understanding is that the model breaks down somewhat in the presence of moisture limitation, which I consider an interesting and valid result/contribution, but you can't have it both ways i.e., the model either does or does not capture fluxes during periods of relative moisture limitation. Perhaps I'm missing

something.

L609: Still need more convincing about how "mountainous" was defined and why these sites were chosen, in particular with respect to other "mountainous" sites in the FLUXNET2015 database. I'm thinking of sites in New Mexico and possibly Oregon off the top of my head.

Chang, L.-L., et al. (2018). Why do large-scale land surface models produce a low ratio of transpiration to evapotranspiration? Journal of Geophysical Research: Atmospheres, 123(17), 9109–9130. https://doi.org/10.1029/2018JD028159

Wieder, W. R., et al. (2017). Ecosystem function in complex mountain terrain: Combining models and long-term observations to advance process-based understanding. Journal of Geophysical Research: Biogeosciences, 122(4), 825–845. https://doi.org/10.1002/2016JG003704

---

## Referee Comment (RC2) · Anonymous Referee #2 · 23 Oct 2020

The topic is very interesting, but the manuscript needs strong improvements and clarifications. Main concerns are on i) Reco, which is included as second variable in the study, while the net ecosystem exchange (NEE) can be more appropriate because it is the actual key term of ecosystem carbon exchanges and it is also directly estimated by eddy covariance based towers (see comment 1), ii) HPM seems interesting but key elements of the model calibration are not provided (e.g., time and spatial scales, parameters), iii) innovative and relevant findings need to be clarified, showed and highlighted.

Detailed comments:

[Figure]

1) Row 34: why Reco? I can't understand why the second variable considered in the study was the ecosystem respiration. It is not observed directly by FLUXNET network, but can be estimated indirectly from net ecosystem exchange (NEE) measurements made by eddy covariance towers during the night. The main term is NEE, why are you not considering it directly? NEE is the key variable considered worldwide. Please, include NEE.

2) Rows 101-105: please, include also SENTINEL 2, the new satellite for NDVI observations with better time and spatial resolutions, available from 2015.

3) Rows 162-165: mean annual precipitation of the watershed is 1200 mm/y. Hence, how can be representative these stations?

4) Rows 194-196: basin area? slope?

5) Row 207: why 16 locations? And not 10 or 20? Please, any sensitivity analysis? Any uncertainty estimate?

6) Equation (1): this equation is the NDVI definition, you don't need to include in the text, it is well known.

7) Row 264: please, include time resolution of the model, its space resolution, and the size of the domain.

8) Row 291: how is estimated g?

9) Row 300: how are estimated Wf, Uf, and bf?

10) Row 318: how many parameters in total?

11) Equation (10) and (11): you don't need to include these equations. These are statistical index very well known.

12) Row 362-363: I looked at section 4.1 and it doesn't estimate any temporal dependency. It just tested the model at a not very clear time scale

13) Row 374: again, what is the time scale?

14) Row 390: is it always at monthly time scale? please, again, define the time scale

15) Row 396-307, "…which also indicates that soil moisture data is necessary to increase Reco prediction accuracy in this ecoregion…": how can you support this statement?

16) Row 415: Are the model parameters changing for each site? What are the parameter values?

17) Row 419-420: I don't agree, Reco predictions are not good in US Whs and US Var

18) Row 518-519, "This result indicates small-scale meteorological forcing and vegetation heterogeneity are the major controls of differences in ET and Reco at the East River Watershed": please, highlight and clarify what is the new finding. We know already that meteorological forcing (which is the model input), and vegetation heterogeneity (model parameter) are the controlling factors of the model.

19) Row 673: please add the journal name of this reference, I can't find it.

---

## Author Comment (AC1) · 29 Nov 2020

We thank the reviewer for spending time to review our manuscript and appreciate all the helpful suggestions and comments. Below, we describe how we have addressed the reviewer's comments in the revised manuscript. .

*Reviewer: This study tested and validated a hybrid predictive modeling (HPM) approach at eight flux tower sites and three snow measurements in western North America. Modeled predictions of annual evapotranspiration and ecosystem respiration fluxes were significantly correlated with observations but less accurate at sub-annual resolution. These results are very promising but also demonstrate some limitations of the current HPM approach.*

*General comments: This is a thorough analysis and very promising study for the application of hybrid models to simulate ecosystem fluxes. It's also a well written manuscript. However, minimal effort has been devoted to generating the type of process-based/transferrable information that's expected from a top journal. Instead, the discussion is generally couched in terms of supporting previous work to emphasis model performance. In my opinion, some re-interpretation of results is needed to move this beyond a basic model validation.*

*A good recent example is Wieder et al. 2017 that also compared CLM (point mode) to flux tower observations in complex terrain. Although their study only considered one site, they explicitly focused on periods of relative agreement versus disagreement between the modeled and observed fluxes to yield broadly testable ecohydrological hypotheses. Given the multi-site focus of the current study, I don't think the same level of detailed inquiry is required, but additional synthetic analysis would increase the scope and the subsequent impact of this work.*

*Relatedly, Figures 4-9 all show similar long-term time series data with scatterplots that lend themselves to similar interpretations in terms of R2 of MAE. These are useful, but perhaps they could be condensed and/or supplemented with other figure types that were more conductive to process-based interpretation. For example, I found Figures 11e and 11f fascinating insofar as they highlighted seasonal differences between vegetation types, but little explanation was provided to "unpack" these results (grasslands and shrublands not even mentioned). Likewise, Figures 12a and 12b present a rich opportunity to speak to differences between the biophysical controls on ET at the SNOTEL and East River sites. Some of the specific factors I'm left wondering about are differences in snow accumulation and melt between sites, evaporation versus transpiration, and heterotrophic versus autotrophic respiration. I understand that you don't have all these measurements, but you've generated a lot of suggestive data that could be leveraged to push this field of research.*

Response: We thank the reviewer for his high-level evaluation and suggestions. We agree with the reviewer that process-based interpretation of watershed dynamics is highly valuable, while such interpretation was present but not adequately described in our initial manuscript. We tried to improve such interpretation in the present version (Also at Q13, Q14 and Q18). Correspondingly, we expanded our discussion on how snowmelt timing influences vegetation dynamics and how different vegetation responds to snowmelt at the East River Watershed with the available NDVI dataset (L481-L491). With the HPM-based estimation results at the East River Watershed, we further unpack how ET and $R_{ECO}$

dynamics shift correspondingly to snowmelt timing, and vary among different vegetation groups, including shrublands and grasslands. We extracted detailed numbers in 2012 (earlier snowmelt year) and 2015 (normal year) for comparison to support our findings and explored how the interactions between snowmelt and vegetation influence ET and $R_{ECO}$ dynamics (L520-L529). We also emphasized what we have observed for shrublands and meadows throughout the result and discussion sections and expanded the discussion around figure 12. In addition, we further clarified the role of meteorological forcing attributes and vegetation types in ET and $R_{ECO}$ dynamics at the East River Watershed and investigated how input variables control ET and $R_{ECO}$ differences among different sites (L535-L550). Following the reviewer's comments, we also clarified the reasons about feature selection for current HPM development (L334-L339) and recommended to include soil moisture or other important parameters for sites with seasonally dry periods (L575-L587). Considering the reviewer's comment about mountainous watersheds, we clarified the reason guiding Fluxnet, SNOTEL and the East River Watershed sites selection (L165-L178). We also included US-Me2 (Oregon site) in this study as suggested in Q21 (Figure A5).

In the following sections, we have addressed the reviewer's comments point-by-point.

Specific Comments:

*Q1: L66-67: Reading back through the manuscript, this seems at odds with the practice of using single flux towers to represent the larger ecoregion (section 4.2). I don't actually have a problem with that research design, but this heterogeneity discussion may not be the best way to set things up.*

Response: We have improved the discussion on using flux towers while trying to evaluate the effect of heterogeneity in the revised manuscript (L66-L73). We agree that flux towers are mainly installed and maintained at relatively flat areas and thus provide fluxes and meteorological conditions that are not indicative of the heterogeneity in each ecoregion. Still, we believe that HPM models trained at these sites can capture the interactions among ET, $R_{ECO}$ and meteorological forcing and vegetation data and then reproduce at some extent the heterogeneity in ET and Reco within the ecoregion based on the variability in meteorological forcing and vegetation data. We validated this approach in the Use Case 2. While the results obtained in the various Uses cases show that the developed approach performed well, they also indicate the current limitations of this approach. The main limitation is due to insufficient resolution of meteorological reanalysis product, which did not reflect the corresponding heterogeneity. In addition, snow and soil moisture data could presumably improve HPM estimation especially during seasonally dry periods, however these datasets are difficult to be obtained over space and time. In the discussion section of the revised manuscript, we have also acknowledged ongoing research that focus on resolving data limitation issues caused by complex terrain, such as NASA's Asteroid Redirect Mission and Surface Atmosphere Integrated Field Laboratory Data.

*Q2: L 73-75: Also uneven hydrologic distribution due to lateral flow in complex terrain (e.g., Chang et al. 2018) that results in heterogeneous fluxes.*

Response: We have now acknowledged this perspective (L83-L86).

*Q3: L109: Has NDVI been defined?*

Response: We have now added the definition (L116).

*Q4: L142-144; L365-367: After reading through the manuscript once, I'm not convinced this objective was met or even really addressed, which was confusing because I kept expecting to come across these results. The small-scale heterogeneity results must be expanded or else it may not be a fatal flaw to just remove this language/objective if the analysis didn't work out (as you intimate on L574-577). In any case,*

*the current manuscript introduction/objectives/results are inconsistent with respect to the degree of focus on this topic.*

Response: We agree that the stated objective of reconstructing small-scale heterogeneity was not entirely met. While the method has shown promising results for predicting ET at other locations in similar topographic position (i.e., flat area), we intended to investigate how meteorological forcing and vegetation heterogeneity influence ET and $R_{ECO}$ at the East River Watershed and the nearby SNOTEL stations. With comparison of meteorological forcing data between weather station and DAYMET data (Figure S3 and S4), we concluded that the insufficient resolution in meteorological reanalysis products limited the ability to estimate the spatiotemporal variability of ET and $R_{ECO}$ in mountainous watersheds, where slope aspect influence the energy balance. This confirm the importance of improving meteorological reanalysis products. While we recognize the above limitation, the impact of vegetation on the ET and $R_{ECO}$ dynamics can be assessed as NDVI data are obtained at much higher resolution. This is what we focused on in section 4.4. The objective and related discussions have been modified and clarified (L148-L151, L366-L374).

*Q5: L143: Replace "CO" with "Colorado, USA" for the global audience.*

Response: We have made this change (L150).

*Q6: L150: I'm curious how you defined "mountainous watersheds" for this study. I've been to the Walnut Gulch sites and they didn't strike me as the least bit mountainous. Also, with respect to my comment on L142-144, how important is the "mountainous" aspect anyway? I understand the broader impacts for water resources, but you'd reach a wider audience if the results were presented in a more general way. I see advantages and disadvantages to both mountain-specific and general analyses, but details/justification (mountain) or else re-framing (general) is needed in either case.*

Response: We define mountainous watersheds as high-elevation watersheds that encompass mountainous ecozones (e.g., montane, subalpine and alpine areas). In this study, we mainly refer to the East River Watershed in Colorado, USA as the representative site for mountainous watersheds (Use Case 4).

The FLUXNET sites were selected to test HPM's capability and limitations under different climate conditions, which may not necessarily locate in mountain regions. For example, US-Ton and the Walnut Gulch site should not be treated as mountainous watershed sites. We have clarified the language and the justification in the revised manuscript (L157-L163, L166-L170).

*Q7: L162: How were the eight FLUXNET stations selected? Some justification needed here. Was it to facilitate the paired approach in section 4.2?*

Response: The Fluxnet sites were selected to sample different climate type, cover a wide range of meteorological and vegetation conditions, and provide continuous >5 years' time series data. These sites represent different ecoregions from Californian Mediterranean, to Sierra Madre Piedmont to Western Cordillera and Boreal Plain. Using sites in various ecosystems enabled us to evaluate the performance of HPM across different sites located in the same ecoregion and evaluate differences in processes driving the ET and $R_{ECO}$ response in various ecoregions. For example, at ecoregions limited by energy conditions (e.g., CA-Oas), current HPM estimations are good, whereas at ecoregions with seasonally dry periods, additional variables (e.g., soil moisture data) might be needed to improve HPM accuracy. The choices of these sites not only facilitated the paired approach in Use Case 2 and 4, but also enabled us to assess HPM limitations at different ecoregions. In the recent revision of the manuscript, additional sets were considered, including US-Me2 (Q21).

*Q8: L164: Table 1 indicates that the Saskatchewan sites are colder than US-NR1.*

Response: We have corrected this mismatch (L168).

*Q9: Table 1: I assume the periods of records are truncated at 2015 because you used the FLUXNET2015 product? This should be specified. Watch significant figures through-out this table.*

Response: Yes. We have clarified this point in the revised manuscript (L190).

*Q10: L227: Why was it necessary to treat this site different than the others? Please provide details about this "cleaning" procedure and why it was needed.*

Response: We have identified some data gaps and erroneous data for the ET data at US-NR1 from the FLUXNET2015 database. The data cleaning framework provided in Rungee et al., 2019 is well documented. We made this decision after visualizing the raw data at US-NR1, where measurements during winter periods are likely uncertain. (L233-L234)

*Q11: L367: The previous text makes it sounds like three (nor four) cases – confusion.*

Response: Thank you for your comment. We have clarified the paragraph (L370-L372).

*Q12: Table 3: You probably don't need a table just to say that "sn" was included at three of the eight sites. Especially because you already have so many display items.*

Response: Thank you for your comments. We have made the recommended change.

*Q13: L378-380: I'm very curious as to whether this was also the case at the seasonally dry Walnut Gulch sites? If so, it speaks to systematic bias where the model captures ET dynamics during energy-limited but not water-limited periods. This strikes me as a major result (see general comments) and could be leveraged to make recommendations about the input variables that are necessary for various systems.*

Response: Thank you for your comments. We agree that the current HPM models with only meteorological attributes and NDVI as features generally captures ET dynamics during energy-limited but not water-limited periods. Variables (i.e., precipitation and constructed index, *sn*) provide indirect information regarding water stress, and we have observed high prediction accuracy during winter time and early growing season. However prediction accuracy usually decreases during peak growing season (summer time), especially at ecoregions that experience dry periods (e.g., occurrence of drought (Sloat et al., 2015; Wainwright et al., 2020). Based on these observations, HPM performs better during energy-limited periods than water-limited periods. As soil moisture is the variable that directly quantifies subsurface moisture stress, including soil moisture as a key variable at HPM can be effective in improving HPM performance. However, soil moisture data highly depends on depth and other subsurface properties (more details provided in Q19). Due to these reasons, we decided not to include soil moisture at the current stage for HPM development (L334-L339) and emphasized the importance of including necessary variables when data becomes available for various systems (L615-L626).

*Q14: L383-395: Wouldn't the model overestimate (not underestimate) Reco if it can't account for soil moisture limitation during this time? Please clarify*

Response: LSTM captures the long-term temporal fluctuations over time really well. But less frequent signals from peak growing season can be neglected due to the decreasing statistical significance. The underestimation of $R_{ECO}$ during peak growing season is resulted from LSTM emphasizing on capturing long term dynamics and smoothing the larger values that occur less frequently.

*Q15: L408-411: Seemingly contradicts L66-67*

Response: We have further clarified this bullet point in the revised manuscript (L66-L73). Also see response for Q1 response.

*Q16: L493: Units mismatch*

Response: We have made the necessary correction (L519).

*Q17: L495-505: Discussion*

Response: We have made the necessary changes.

*Q18: L516-518: In my mind, this is a missed opportunity to gain processed-based (and thus transferrable) insight. What about these sites could factor into ET differences that are so much greater than the Reco differences? See general comments.*

Response: Thank you for your comment. Through further investigating data inputs and model performance, we believe the ET and $R_{ECO}$ estimation at the East River Watershed are limited by the insufficient resolution of input data. There are two major factors that lead to the differences of ET and $R_{ECO}$ among these sites. The first perspective is that HPM ET model is more sensitive to temperature and radiation inputs compared to NDVI whereas NDVI, temperature and radiation are all influential for $R_{ECO}$ estimations. The second perspective is about data resolution and uncertainty. Data provided by SNOTEL weather stations are more accurate than DAYMET reanalysis data. We observed a greater differences in temperature and radiation at the SNOTEL sites whereas there's very small differences at the East River sites (Figure S3). Summer temperature differences among SNOTEL sites can be over 3℃ but there's a barely 0.2℃ differences in DAYMET data used for the East River sites. From Landsat data, we could distinguish the differences in NDVI at the East River sites (Figure 10) and these differences are well captured by HPM $R_{ECO}$ model. Considering both the model and data perspectives, it explained why we observed these differences in ET and $R_{ECO}$ at these sites. With the high prediction accuracy for Use Case I scenarios, we believe HPM is capable to capture the interactions among ET, $R_{ECO}$ and input variables. If high resolution meteorological data becomes available at the East River Watershed such as the Surface Atmosphere Integrated Field Laboratory (SAIL), we believe HPM can better distinguish how meteorological forcing heterogeneity controls ET and $R_{ECO}$ and more process-based interpretation can be learned from HPM estimations. These discussions have been implemented in the revised manuscript (e.g., L535-L550; L597-L604; L627-L637).

*Q19: L544-546: How hard would it be to add moisture into the model? Why wasn't it added in the first place? I'm not suggesting that you re-do the analysis, but the readers will be very interested in this information.*

Response: For Use Case 1 situations, it is applicable and relatively easy to add soil moisture into the model. However, for the other Use Cases where the model will be transferred over space, it is difficult to directly use soil moisture as an input variable given soil moisture measurements are often made at various depths, and other relevant  soil characteristics are likely  different at different sites. The dependence on depth and other soil characteristics limits the model transferability at inadequately monitored watersheds (L334-L339). Thus, we have decided not to include soil moisture as an input variable. In L615-L626 of the revised manuscript, we now recommend the inclusion of soil moisture and other important variables if data are available and researchers have information regarding site characteristics. For example, at ecoregions that experience seasonally dry periods, it is useful to add soil moisture in HPM.

*Q20: L563-568: It's not clear to me what model results "present similar dynamic trends" to the moisture limitation invoked by Hu et al. 2010 (and a host of larger scale, more recent work). My current understanding is that the model breaks down somewhat in the presence of moisture limitation, which I consider an interesting and valid result/contribution, but you can't have it both ways i.e., the model either does or does not capture fluxes during periods of relative moisture limitation. Perhaps I'm missing something.*

Response: Thank you for your comment. We meant to say ET and $R_{ECO}$ estimation from HPM at the East River Watershed are comparable to the other studies. Specifically, Hu et al. (2010) were able to use snow, branch and soil sample data at US-NR1 to conclude that a longer growing season led to less carbon sequestration. At the East River Watershed, HPM estimated smaller $R_{ECO}$ for evergreen forests (592 to $639 gCm^{-2}$) that have longer growing season, compared to deciduous forests (642 to 698 $gCm^{-2}$). In the revised manuscript, we also compared 2012 (earlier snowmelt and longer growing season length) to other years and discussed how these dynamics influence seasonal and annual ET and $R_{ECO}$ (L520-529).

The current HPM model does not include soil moisture due to reasons described in response to Q19, however other attributes, such as precipitation and sn, provide indirect information regarding the moisture inputs. Thus, fluxes estimated from HPM during moisture limiting periods are still reasonable. In the revised manuscript, we have demonstrated this limitation of HPM and suggested to incorporate soil moisture data in addition to precipitation and sn when they become available.

*Q21: L609: Still need more convincing about how "mountainous" was defined and why these sites were chosen, in particular with respect to other "mountainous" sites in the FLUXNET2015 database. I'm thinking of sites in New Mexico and possibly Oregon off the top of my head.*

Response: Thank you for your comment. In our study, the representative site of mountainous watersheds is the East River Watershed, and the surrounding US-NR1 and SNOTEL sites. Other Fluxnet sites were selected to test the capability and limitations of HPM under other climate conditions (e.g., US-Ton, US-Wkg) and are not considered as mountainous sites. We have also included US-Me2 (Oregon) in our study. We did not include US-VCM (New Mexico site) after recognizing the occurrence of fire in 2013. These changes have been clarified in the revised manuscript. We have also added the estimation of ET and $R_{ECO}$ at US-Me2. Results of US-Me2 is attached here and also included as Figure A5 in the revised manuscript.

[Figure]

Figure 1. Added HPM estimation of ET and $R_{ECO}$ at US-Me2. Data from 2011 are partially missing, which may decreases LSTM performance at US-Me2.

Reference:

Sloat, L. L., Henderson, A. N., Lamanna, C. and Enquist, B. J.: The Effect of the Foresummer Drought on Carbon Exchange in Subalpine Meadows, Ecosystems, 18(3), 533–545, doi:10.1007/s10021-015-9845-1, 2015.

Wainwright, H. M., Steefel, C., Trutner, S. D., Henderson, A. N., Nikolopoulos, E. I., Wilmer, C. F., Chadwick, K. D., Falco, N., Schaettle, K. B., Brown, J. B., Steltzer, H., Williams, K. H., Hubbard, S. and Enquist, B. J.: Satellite-derived foresummer drought sensitivity of plant productivity in Rocky Mountain headwater catchments: spatial heterogeneity and geological-geomorphological control, Environ. Res. Lett., doi:10.1088/1748-9326/ab8fd0, 2020.

---

## Author Comment (AC2) · 29 Nov 2020

We appreciate the reviewer's efforts in reading our manuscript and providing useful comments and recommendations. In this document, we provide answers to reviewer's questions and explain how we have modified and improved our manuscript.

*Reviewer general comment: the topic is very interesting, but the manuscript needs strong improvements and clarifications. Main concerns are on i) Reco, which is included as second variable in the study, while the net ecosystem exchange (NEE) can be more appropriate because it is the actual key term of ecosystem carbon exchanges and it is also directly estimated by eddy covariance based towers (see comment 1), ii) HPM seems interesting but key elements of the model calibration are not provided (e.g., time and spatial scales, parameters), iii) innovative and relevant findings need to be clarified, showed and highlighted.*

Response: We made a concerted effort to incorporate the reviewer's suggestions into the revised version of the manuscript. With regard to replacing $R_{ECO}$ by NEE in our study (reviewer's concern #1), we agree that net ecosystem exchange (NEE) is a key variable considered worldwide, and eddy covariance towers directly measure NEE. However, $R_{ECO}$ is also a very important parameter recognized by researchers (Le Quéré et al., 2009). In this study, we decided to concentrate on estimating $R_{ECO}$, and more thorough response on the reasons can be found in Q1. With regard to describing the key elements of the model calibration (reviewer's concern (ii)), we clarified the configuration of the deep learning module we used in addition to the information that was provided in table 1 in the supplementary material. We have also demonstrated how we selected parameters used for HPM. Related responses can be found at Q6, Q7, Q8, Q9 and Q15. Finally, with regard to innovative and relevant findings needing clarification (reviewer's concern (iii)), we expanded the discussion on providing more process-based interpretation of watershed dynamics and highlighted the relevant innovative results. With the guidance of the reviewer, we think the quality of the manuscript has been improved.

Specific Comments:

*Q1: Row 34: why Reco? I can't understand why the second variable considered in the study was the ecosystem respiration. It is not observed directly by FLUXNET network, but can be estimated indirectly from net ecosystem exchange (NEE) measurements made by eddy covariance towers during the night. The main term is NEE, why are you not considering it directly? NEE is the key variable considered world wide. Please, include NEE.*

Response: We agree that NEE is one of the key variables considered worldwide, and is directly measured by eddy covariance flux towers. However, $R_{ECO}$ is also very important as it represents the total ecosystem carbon emissions from land to the atmosphere, and is very sensitive to climate change (Le Quéré et al., 2009), and thus quantifying and estimating $R_{ECO}$ is needed. This study is not the only one that concentrates on $R_{ECO}$. For example, Ai et al. (2018) developed a semi-empirical, physiologically based, remote sensing model to estimate $R_{ECO}$ using MODIS data; Solomon et al. (2013) estimated daily respiration rates using maximum likelihood fits of a free-water metabolism to quantify respiration dynamics in six lakes.

An additional reason to consider $R_{ECO}$ and not NEE in this study is that one of the major objectives of this study is to provide an estimate of ET and $R_{ECO}$ at watersheds where flux towers are not available. The

daytime and nighttime partitioning methods (van Gorsel et al., 2009; Reichstein et al., 2005) requires sub-daily scale NEE data to compute daily scale $R_{ECO}$. However at these sparsely monitored watersheds, sub-daily scale NEE data is not available and could not be predicted with weather reanalysis and remote sensing data that are at coarser temporal scales. Thus developing methods that estimate daily scale $R_{ECO}$ is still needed and will help advance our understanding of ecosystem dynamics and carbon cycling at the inadequately monitored watersheds.

While we decided to not include NEE in our manuscript, we have tested the HPM approach to estimate NEE at CA-OAS and US-NR1. We observed a $R^2$ larger than 0.8 between the measurements and predictions (Figure 1). With this result, we believe HPM can be an appropriate approach for estimating daily NEE with right choices of variables. However, we believe replacing $R_{ECO}$ with NEE will change the scope of this study, and thus we do not plan to include NEE at the current stage.

[Figure]

Figure 1. HPM estimate of NEE at CA-OAS and US-NR1. $R^2$ between estimation and measurements are 0.87, 0.83 and 0.81 at CA-OAS; 0.94, 0.88 and 0.90 at US-NR1 for the training set, validation set and prediction set, respectively. Model inputs include air temperature, soil temperature, sn, precipitation and radiation.

*Q2: Rows 101-105: please, include also SENTINEL 2, the new satellite for NDVI observations with better time and spatial resolutions, available from 2015.*

Response: We considered using SENTINEL 2 data but our evaluation has shown that LANDSAT was more adapted for the period of time we are concentrating on in this study. From our knowledge, the Sentinel 2 surface reflectance data has a 10m resolution for the red and near infrared band with an averaged revisit time of 5 days since March 2017 (Main-Knorn et al., 2017). In Use Case I, II and III scenarios, our HPM estimates covered the period up to 2016 as we were using FLUXNET2015 datasets. For Use Case IV, we have checked the data availability of Sentinel 2 surface reflectance data over the East River Watershed. A total of 106 Sentinel 2 dataset is available till 2018, however only 13 of them have a cloud cover less than 10% during the sampling period. We tested with the additional Sentinel-2 NDVI data, but we did not observe notable changes in ET and $R_{ECO}$ estimations compared to previous ones with only Landsat data. Meanwhile, we also checked out other satellite products, including the Planet-Lab (McCabe et al., 2017); the harmonized Landsat-Sentinel product (Claverie et al., 2018) as well as other satellite data fusion products (Shao et al., 2019). However, they still do not increase the temporal resolution prior to 2017. Based on the above assessment we decided to use Landsat data only. We agree

that future work could expand HPM based ET and $R_{ECO}$ estimations using various combination of satellite products.

*Q3: Rows 162-165: mean annual precipitation of the watershed is 1200 mm/y. Hence, how can be representative these stations?*

Response: We agree the manuscript did not clearly describe the reason of using FLUXNET site with very different meteorological forcing from the East River watershed. We have improved the manuscript to make it clearer. The FLUXNET sites considered in this study is mainly used in Use Case I and Use Case II. We wanted to explore the capability of HPM under different climate conditions. For example, we selected US-Ton to test whether HPM is able to provide reasonable estimate of ET and $R_{ECO}$ under Mediterrean climate (Csa) whereas at US-NR1 for subarctic environment (Dfc). We did not intend to use HPM developed at US-Ton and other FLUXNET sites to be representative stations for the East River Watershed. We have further clarified the major objective of and the various Use Cases in the revised manuscript (L157-L167).

*Q4: Rows 194-196: basin areas? slope?*

Response: We were describing the general characteristic of the East River Watershed, which include both basin areas and montane areas.

*Q5: Rows 207: why 16 locations? And not 10 or 20? Please, any sensitivity analysis? Any uncertainty estimate?*

Response: We focused on four main vegetation types within the East River Watershed, including deciduous forests, evergreen forests, riparian shrublands and meadow grasslands. We defined 16 locations to have some replicates to evaluate the spatial variability. Given the 30-m spatial resolution of Landsat, we tried to select locations at the center of vegetation patched and covered or at least strongly dominated by one vegetation type. We evaluated it manually and decided that 16 locations (4 for each vegetation type) was a pragmatic choice.

*Equation (1): this equation is the NDVI definition, you don't need to include in the text, it is well known.*

Response: We have made the necessary change in the revised manuscript.

*Q6: Row 264: please, include the time resolution of the model, its space resolution, and the size of the domain.*

Response: The time resolution of HPM is daily. The spatial resolution depends on different Use Cases and spatial resolution of data inputs. We have made these changes in the revised manuscript, and also further clarified in each of the Use Cases in section 4 (L337-L338).

*Q7: Row 291: how is estimated g?*

Response: *g* is the hyperbolic tangent activation function. It is used to determine candidate cell states and update the hidden states (Hochreiter and Schmidhuber, 1997; Kratzert et al., 2019). g is not estimated.

*Q8: Row 300: how are estimated Wf, Uf, and bf?*

Response: We clarified this. $W_f, U_f$ and $b_f$ are representing learnable parameters for the forget gates. There are other $W, U$ and $b$ for the internal states and hidden states. We used the Adam algorithm (Kingma and Ba, 2014) with a mean absolute error loss function built in Keras (Chollet, 2015).

*Q9: Row 318: how many parameters in total?*

Response: We thank the reviewer's comment. There are 11600 and 7600 parameters for the first and second LSTM layers; 208 and 9 for the first and second dense layers and no parameters for the dropout layers. These information is available in Table 1 in the supplementary material.

*Q10: Equations (10) and (11): you don't need to include these equations. These are statistical index very well known.*

Response: We have made the necessary change in the revised manuscript.

*Q11: Row 362-363: I looked at section 4.1 and it doesn't estimate any temporal dependency. It just tested the model at a not very clear time scale.*

Response: We used the word "long term temporal dependency" in a deep learning (or statistical) context where there are significant temporal correlation and long time lags in time series. LSTM and recurrent neural networks are one of the very efficient and effective deep learning models that are capable to capture such long term dependencies. In the revised manuscript, we have been more careful in terms of the wording that overlap among different fields.

*Q12: Row 374: again, what is the time scale?*

Response: We thank the reviewer's comment. Our ET and $R_{ECO}$ estimations are at daily scale. We have better clarified the time scale in the manuscript.

*Q13: Row 390: Is it always at monthly time scale? please, again, define the time scale.*

Response: We thank the reviewer's comment. Our estimate is at daily scale, which are used to calculate monthly means. In figure 5 and following figures, we presented the results of both daily and monthly mean between HPM and measurements in order to check model performance. We also aggregated our daily estimates to 8-day mean at the East River sites, which enabled us to compare our results to Mu et al. (2013) as shown in Figure S1

*Q14: Row 396-307, '...which also indicates that soil moisture data is necessary to increase Reco prediction accuracy in this ecoregion...". how can you support this statement?*

Response: We thank the reviewer's comment. The decreasing prediction accuracy occurred at sites limited by water and moisture conditions (e.g., US-Ton and US-Var) where other studies investigated how ecosystem respond to subsurface water availability (Von Buttlar et al., 2018; Song et al., 2014). At sites with seasonally dry periods (e.g., US-NR1), other studies have also identified the occurrences of fore-summer drought and water limiting condition (Sloat et al., 2015; Wainwright et al., 2020). Due to practical reasons, current HPM models did not include soil moisture as an input to capture these seasonally dry periods. Thus we believe modified HPM models with soil moisture as inputs can increase prediction accuracy in these ecoregions when soil moisture data becomes more available in space and time.

*Q15: Row 415: Are the model parameters changing for each site? What are the parameter values?*

Response: We thank the reviewer's comment. Deep learning parameters for US-Ton, CA-Oas and US-Wkg are different as they are three different developed HPM models used to represent different ecoregions. As mentioned in the response to Q9, there are many deep learning parameters and it is not feasible to directly present the values of these learnable parameters here. But all of these parameter values

are available in the data package we submitted to ESS-DIVE and can be downloaded at https://data.ess-dive.lbl.gov/view/doi:10.15485/1633810 named 'LSTM_model.zip'.

*Q16: Row 419-420: I don't agree, Reco predictions are not good in US Whs and US Var*

Response: We thank the reviewer's comment. HPM achieved a daily scale adjusted $R^2$ of 0.70 and 0.78 and $MAE$ at 0.67 and 0.22 at US-Whs and US-Var respectively, in Use Case II scenario. We agree that the statistical measure is not as satisfactory as $R^2$ over 0.9. In the revised manuscript, we have made the necessary changes correspondingly (L433-L435).

*Q17: Row 518-519, "This result indicates small-scale meteorological forcing and vegetation heterogeneity are the major controls of differences in ET and Reco at the East River Watershed": please, highlight and clarify what is the new finding. We know already that meteorological forcing (which is the model input), and vegetation heterogeneity (model parameter) are the controlling factors of the model.*

Response: We thank the reviewer for the comment. We have expanded our discussion on how ET and $R_{ECO}$ dynamics vary at different years (e.g., years with earlier snowmelt versus later snowmelt). We have emphasized how vegetation types contribute to ET and $R_{ECO}$ spatiotemporal heterogeneities. We also discussed the limitations and practical perspectives of current HPM models in feature selection and how to improve estimation accuracy at seasonally dry periods. In addition, we clarified the role of meteorological forcing attributes and vegetation types in ET and $R_{ECO}$ dynamics at the East River Watershed and tested how these input variables contribute to ET and $R_{ECO}$ differences at different years among different sites. These findings have been revised in the result and discussion sections.

*Q18: Row 673: please add the journal name of this reference, I can't find it.*

Response: We thank the reviewer's comment. The journal name of this reference is '*Journal of Geophysical Research: Biogeosciences*' (L734-L736). We have double checked and made sure the bibliography is correctly and clearly presented.

Reference:

Ai, J., Jia, G., Epstein, H. E., Wang, H., Zhang, A. and Hu, Y.: MODIS-Based Estimates of Global Terrestrial Ecosystem Respiration, J. Geophys. Res. Biogeosciences, 123(2), 326–352, doi:10.1002/2017JG004107, 2018.

Von Buttlar, J., Zscheischler, J., Rammig, A., Sippel, S., Reichstein, M., Knohl, A., Jung, M., Menzer, O., Altaf Arain, M., Buchmann, N., Cescatti, A., Gianelle, D., Kiely, G., Law, B. E., Magliulo, V., Margolis, H., McCaughey, H., Merbold, L., Migliavacca, M., Montagnani, L., Oechel, W., Pavelka, M., Peichl, M., Rambal, S., Raschi, A., Scott, R. L., Vaccari, F. P., Van Gorsel, E., Varlagin, A., Wohlfahrt, G. and Mahecha, M. D.: Impacts of droughts and extreme-temperature events on gross primary production and ecosystem respiration: A systematic assessment across ecosystems and climate zones, Biogeosciences, doi:10.5194/bg-15-1293-2018, 2018.

Chollet, F.: Keras: The Python Deep Learning library, Keras.Io, 2015.

Claverie, M., Ju, J., Masek, J. G., Dungan, J. L., Vermote, E. F., Roger, J. C., Skakun, S. V. and Justice, C.: The Harmonized Landsat and Sentinel-2 surface reflectance data set, Remote Sens. Environ., doi:10.1016/j.rse.2018.09.002, 2018.

van Gorsel, E., Delpierre, N., Leuning, R., Black, A., Munger, J. W., Wofsy, S., Aubinet, M.,

Feigenwinter, C., Beringer, J., Bonal, D., Chen, B., Chen, J., Clement, R., Davis, K. J., Desai, A. R., Dragoni, D., Etzold, S., Grünwald, T., Gu, L., Heinesch, B., Hutyra, L. R., Jans, W. W. P., Kutsch, W., Law, B. E., Leclerc, M. Y., Mammarella, I., Montagnani, L., Noormets, A., Rebmann, C. and Wharton, S.: Estimating nocturnal ecosystem respiration from the vertical turbulent flux and change in storage of CO 2, Agric. For. Meteorol., 149(11), 1919–1930, doi:10.1016/j.agrformet.2009.06.020, 2009.

Hochreiter, S. and Schmidhuber, J.: Long Short-Term Memory, Neural Comput., doi:10.1162/neco.1997.9.8.1735, 1997.

Kingma, D. P. and Ba, J.: Adam: A Method for Stochastic Optimization, , 1–15 [online] Available from: http://arxiv.org/abs/1412.6980, 2014.

Kratzert, F., Klotz, D., Herrnegger, M., Sampson, A. K., Hochreiter, S. and Nearing, G. S.: Toward Improved Predictions in Ungauged Basins: Exploiting the Power of Machine Learning, Water Resour. Res., doi:10.1029/2019WR026065, 2019.

Main-Knorn, M., Pflug, B., Louis, J., Debaecker, V., Müller-Wilm, U. and Gascon, F.: Sen2Cor for Sentinel-2., 2017.

McCabe, M. F., Aragon, B., Houborg, R. and Mascaro, J.: CubeSats in Hydrology: Ultrahigh-Resolution Insights Into Vegetation Dynamics and Terrestrial Evaporation, Water Resour. Res., 53(12), 10017–10024, doi:10.1002/2017WR022240, 2017.

Mu, Q., Zhao, M. and Running, S. W.: MODIS Global Terrestrial Evapotranspiration (ET) Product (MOD16A2/A3), Algorithm Theor. Basis Doc., 2013.

Le Quéré, C., Raupach, M. R., Canadell, J. G., Marland, G., Bopp, L., Ciais, P., Conway, T. J., Doney, S. C., Feely, R. A., Foster, P., Friedlingstein, P., Gurney, K., Houghton, R. A., House, J. I., Huntingford, C., Levy, P. E., Lomas, M. R., Majkut, J., Metzl, N., Ometto, J. P., Peters, G. P., Prentice, I. C., Randerson, J. T., Running, S. W., Sarmiento, J. L., Schuster, U., Sitch, S., Takahashi, T., Viovy, N., Van Der Werf, G. R. and Woodward, F. I.: Trends in the sources and sinks of carbon dioxide, Nat. Geosci., doi:10.1038/ngeo689, 2009.

Reichstein, M., Falge, E., Baldocchi, D., Papale, D., Aubinet, M., Berbigier, P., Bernhofer, C., Buchmann, N., Gilmanov, T., Granier, A., Grünwald, T., Havránková, K., Ilvesniemi, H., Janous, D., Knohl, A., Laurila, T., Lohila, A., Loustau, D., Matteucci, G., Meyers, T., Miglietta, F., Ourcival, J. M., Pumpanen, J., Rambal, S., Rotenberg, E., Sanz, M., Tenhunen, J., Seufert, G., Vaccari, F., Vesala, T., Yakir, D. and Valentini, R.: On the separation of net ecosystem exchange into assimilation and ecosystem respiration: Review and improved algorithm, Glob. Chang. Biol., doi:10.1111/j.1365-2486.2005.001002.x, 2005.

Shao, Z., Cai, J., Fu, P., Hu, L. and Liu, T.: Deep learning-based fusion of Landsat-8 and Sentinel-2 images for a harmonized surface reflectance product, Remote Sens. Environ., doi:10.1016/j.rse.2019.111425, 2019.

Sloat, L. L., Henderson, A. N., Lamanna, C. and Enquist, B. J.: The Effect of the Foresummer Drought on Carbon Exchange in Subalpine Meadows, Ecosystems, 18(3), 533–545, doi:10.1007/s10021-015-9845-1, 2015.

Solomon, C. T., Bruesewitz, D. A., Richardson, D. C., Rose, K. C., Van de Bogert, M. C., Hanson, P. C., Kratz, T. K., Larget, B., Adrian, R., Leroux Babin, B., Chiu, C. Y., Hamilton, D. P., Gaiser, E. E., Hendricks, S., Istvá, V., Laas, A., O'Donnell, D. M., Pace, M. L., Ryder, E., Staehr, P. A., Torgersen, T., Vanni, M. J., Weathers, K. C. and Zhu, G.: Ecosystem respiration: Drivers of daily variability and background respiration in lakes around the globe, Limnol. Oceanogr., doi:10.4319/lo.2013.58.3.0849, 2013.

Song, B., Niu, S., Luo, R., Luo, Y., Chen, J., Yu, G., Olejnik, J., Wohlfahrt, G., Kiely, G., Noormets, A., Montagnani, L., Cescatti, A., Magliulo, V., Law, B. E., Lund, M., Varlagin, A., Raschi, A., Peichl, M., Nilsson, M. B. and Merbold, L.: Divergent apparent temperature sensitivity of terrestrial ecosystem respiration, J. Plant Ecol., doi:10.1093/jpe/rtu014, 2014.

Wainwright, H. M., Steefel, C., Trutner, S. D., Henderson, A. N., Nikolopoulos, E. I., Wilmer, C. F., Chadwick, K. D., Falco, N., Schaettle, K. B., Brown, J. B., Steltzer, H., Williams, K. H., Hubbard, S. and Enquist, B. J.: Satellite-derived foresummer drought sensitivity of plant productivity in Rocky Mountain headwater catchments: spatial heterogeneity and geological-geomorphological control, Environ. Res. Lett., doi:10.1088/1748-9326/ab8fd0, 2020.

---

## Referee Report (RR1)

**Review of "hess-2020-322"**
**"A Deep-Learning Hybrid-Predictive-Modeling Approach for Estimating Evapotranspiration and Ecosystem Respiration"**
**Chen et al.**

Anonymous

March 21, 2021

**Meta**

- Disclaimer: I consider myself capable of evaluating the methodological aspects of this study, but I am not an ecologist.

- Note that the list item numbers (e.g., [1-3]) refer to lines in the manuscript.

- Overall great work, thanks for sharing it!

**Summary of the Content**

The study tests a hybrid modeling approach which integrates different data sources using deep learning to predict daily evapotranspiration (ET) and ecosystem respiration ($R_{ECO}$). The authors use eddy covariance FLUXNET site-level data and simulations from a physically-based model to train their model on a set of meteorological features (e.g., precipitation, air temperature) and remotely sensed data (NDVI). The model was tuned on different data to test a range of use cases: Training and testing the HPM 1) within FLUXNET site, 2) between FLUXNET site (genrealizeability), 3) emulating a physically-based model, and finally, to 4) study the connection of ET and $R_{ECO}$ and meteorological forcings and vegetation properties. The model was able to learn patterns of ET and $R_{ECO}$ to a satisfying degree, and connections between ET and $R_{ECO}$ and meteorological forcings and vegetation dynamics have been found to be significant at high spatial resolution.

**Overall Feedback**

Integrating different datasets for ecosystem modeling with neural networks is a hot topic which has the potential to improve the predictability of such complex systems. Much effort has been spent to process different datasets and the proposed approach has been tested in several experiments. I am sure that this study has been conducted with care and I find the results very interesting. I especially like that the authors considered overfitting by using training, validation and prediction sets and tested between-sites genrealizeability.

**Recommendation**

*Minor revisions*: I think the manuscript needs improvement (structure, language, consistency, figures, tables), but the study is still interesting and nice overall.

**Major Remarks**

- I read parts of the manuscript several times to understand how the FLUXNET and CLM data was used (, separately) and how the framawork exactly works, and I am still not sure if I entirely understand it. Also, it took me some time to understand the four experiments ("use cases"), what data was used for training, testing, etc. This is my major critic: I think the manuscript needs a cleaner structure and language.

- For me, the term "hybrid" is a bit confusing here. I assume that you refer to Reichstein (2019), where "(5) Surrogate modelling or emulation" is listed as a hybrid approach, which, once trained, can "achieve simulations orders of magnitude faster than the original physical model without sacrificing much accuracy" and "allows for fast sensitivity analysis, model parameter calibration, and derivation of confidence intervals for the estimates". I think the manuscript would be much easyer to understand if you would make this clearer.

**Minor Remarks**

**General**

- I strongly recommend to use colorblind-friendly colors in the plots. The time-series plots with green and red color mixed are particularly problematic. I think that the figures need some more work (general appearance, font size).

- From the HESS guidelines: "Common Latin phrases are not italicized (for example, et al., cf., e.g., a priori, in situ, [...])" (e.g., line 49, in situ).

- From the HESS guidelines: "The abbreviation "Fig." should be used when it appears in running text and should be followed by a number unless it comes at the beginning of a sentence, e.g.: "The results are depicted in Fig. 5. Figure 9 reveals that...".".

- From the HESS guidelines: "Units must be written exponentially (e.g. W m$^{-2}$).", e.g. line 380 or in axes labels, you use mm/d instead of m d$^{-1}$.

- You use the notation "Adj.R2-0.94" in some figures (e.g. Fig. 5). This is misleading, please use "Adj.R2: 0.94", "Adj.R2=0.94", or similar.

- Time-series figures: please add a legend for all plots (pink points, red, greeen, blue, black lines).

- Symbol notation: I noticed you use "ET" for evapotranspiration and "$R_{ECO}$" for ecosystem respiration. I find this is inconsistent, as either you use these as abbreviations, which are not italic ("ET" & "R$_{ECO}$"), or as mathematical variables ("$E$" & "$R_{ECO}$"), where multi-letter symbols are to be avoided due to ambiguity (is "$ET = E \cdot T$?"), and subsctipts are only italic if they refere to a variable (such as in $x_i$, where $i$ is an index), but not if the subscript is a name.

- In general, many small "not so nice" things like units written inconsistently.

- I suggest to not put "learn" in quotes (as in *the model "learns"*) as the term is very commonly used in this context.

- Nice that you split the data in training, validation, and test (prediction) set! This is often not done.

**Abstract**

- The abstract is too detailed in my opinion, consider to shorten.

- I suggest to state clearly how the approach is hybrid and why you use the approach.

**1. Introduction**

- Nice review of current methods to estimate ET and R$_{ECO}$. It *could* be shortened a bit.

**2. Site Information, Data Acquisition and Processing**

Tab. 1 It is hard to differentiate between the rows visually.

Fig. 1 Consider highlighting the SNOTEL sites visually.

**3. Hybrid Predictive Modeling Framework**

- I think you don't need to explain the LSTM in detail.

260 Does "deeply connected neural networks" refer to a fully connected neural network?

- For *use case 2*, do you train the model on all sates jointly or on single sites?

282-320 Consider replacing the extensive description of LSTMs with a conceptual high-level description.

326 Would be nice to see if a smaller model does the job (but not essential here).

331 Olah. (2015) → Olah (2015)

340-352 Why did you separate precipitation into rainfall and snowfall and how was the variable *sn* used? If they were used as inputs for the LSTM, why not letting the neural network figure this out, i.e., just inputting the avalable features?

355 I assume you used an LSTM? Then you can just use the term LSTM here, as it has been introduced already instead of "deep-learning recurrent neural networks".

**4. Results**

- I sugest to move the descriptions of the "use cases" to the methods section, maybe make a table that summarized what data is used for training and testing, the objective of the experiment etc.

- The interpretation would be much easyer if you would show the mean seasonal cycle and the interannual variability!

399-407 This is already discussion of the results.

399-404 I would expect that the LSTM learns SM dynamic, i.e., it represents it (implicitely) in its hidden state. SM would not necessarily be needed as the LSTM learns the ecological memory effects (e.g., Besnard et al. (2019) or Kraft et al. (2019). Adding SM could still help improving the model as it currently does not have much data to learn from compared to the number of parameters. Also, referring to a comment from former Referee #1, I think this should be clarified. This is one of the key advanteges of using models like an LSTM, it can learn ecological memory and thus, variables such as soil moisture may not be needed!

405-407 I agree that LSTMs tend to have issues with extreme values. In my oppinion, this is mostly because extreme values are rare, i.e., the model does not see many anomalous samples, there is less training data for such cases. Maybe you could mention this and provide a source, if you can find one.

Tab. 3 Please write units in exponential form. You could mention that the increase in test performance could be linked to droupout (which I assume is deactivated for inference) in the discussion.

420 I think the representativity of FLUXNET sites for the entire ecoregion is questionable and disputed (?), maybe rephrase.

Fig. 7 The monthly errors used to be pink before, right? I suggest to resue the same colors.

450 I don't know what an "1-D" model is, consider explaining.

475 The *mechanistic* HPM model?

479 30m → 30 m

479+ Much of it is discussion.

516 17% → 17 %

Fig. 11 Panels (a) and (b) are not very informative, maybe remove?

**5. Discussion**

559 You refered to "physically-model-based HPM" as "mechanistic HPM" (line 264), you may use the latter one here.

625 Again, I think you need to discuss the "memory aspect". If you have meteorologicla data and site-level variables (e.g. vegetation type, soil properties), and enough training data, an LSTM would learn SM implicitely. This should be added to the discussion, as it is a key selling-point for using deep learning models. I think the message "SM is needed for improving model" is wrong, state variables are *not* needed anymore with DL approaches if the states can be derived from the input data. Of course, it can still be beneficial to add soil moisture, as it would regularize the model and maybe, the complex processes involved (e.g., lateral flux) may not be learnt by the model if the relevant features are missing.

651-660 As an outlook: the model could be trained on FLUXNET and process-based simulations *jointly*.

669 I cite reviewer #1: "Replace *CO* with *Colorado, USA* for the global audience." ;)

---

## Author Response (AR2)

Author responses to reviewers' comments

We appreciate the reviewers for providing us constructive comments and suggestions. Here we provide a summary of how we addressed each reviewer's questions or comments. All reviewers' comments and questions are in italic, followed by our detailed responses. We used Lx-Ly to represent the lines associated with changes we made in the revised manuscript and Linex-Liney to refer to specific reviewers' comments.

Summary:

1. We improved the manuscript organization and reduced the length significantly (47 pages to 34 pages) as suggested by the reviewers.

2. We reformatted the figures, adopting more friendly color schemes and using consistent color palette as suggested by reviewer #3.

3. We added relevant references as suggested by the reviewers.

4. We checked with HESS guideline and ensured units and Latin words are correctly written.

5. We revised the methodology section as suggested by reviewer #3. We replaced the detailed mathematical demonstration of LSTM with short but high-level explanation. We also added a sub-section explaining the use cases.

6. We addressed reviewer #1's suggestions to consider snow and monsoon's impacts over fluxes. HPM only requires 4 or 5 input variables, so it does not have the capability to explicitly track the movement of snow water and/or monsoon water. To address reviewer's hypothesis, we separated annual ET and $R_{eco}$ into pre-June (January – June) component and post-July (July – December) component. We also included additional years (e.g., 2014) to have different combinations of snow precipitation and monsoon precipitation.

7. We partly addressed reviewer #1's suggestions to perform additional research on delineating fore-summer drought and post-monsoon droughts; identifying differences in snow-dominated watersheds versus monsoon-dominated watershed and quantifying evaporation versus transpiration and autotrophic versus heterotrophic respiration. We performed an independent analysis based upon the Palmer drought sensitivity index (PDSI) and radiation and precipitation data at the East River Watershed to determine major control of watershed dynamics over time. The results indicated that there were no significant differences in meteorological control between US-NR1 and the East River Watershed that occurrences of fore-summer droughts and post-monsoon droughts are highly correlated, and energy limiting conditions may exert more control on watershed dynamics than moisture limitation during late summer and autumn periods. Details of the analysis can be found in our response to reviewer #1's below. We did not consider evaporation versus transpiration and autotrophic versus heterotrophic respiration as suggested by the reviewer because we do not have data to enable this, and collecting them is beyond the context of this study.

**Our responses to RC#1.**

General comments:

*1. The first paragraphs of my previous review (general comments) were not specifically addressed. In my experience, this is not typical for the reviewer response process, and is likely to be unacceptable to some reviewers and/or editors going forward. In particular, I*

*would still appreciate the authors responding to the third paragraph of my previous review, repeated here:*

*Figures 4-9 all show similar long-term time series data with scatterplots that lend themselves to similar interpretations in terms of R2 of MAE. These are useful, but perhaps they could be condensed and/or supplemented with other figure types that were more conductive to process-based interpretation. For example, I found Figures 11e and 11f fascinating insofar as they highlighted seasonal differences between vegetation types, but little explanation was provided to "unpack" these results (grasslands and shrublands not even mentioned). Likewise, Figures 12a and 12b present a rich opportunity to speak to differences between the biophysical controls on ET at the SNOTEL and East River sites. Some of the specific factors I'm left wondering about are differences in snow accumulation and melt between sites, evaporation versus transpiration, and heterotrophic versus autotrophic respiration. I understand that you don't have all these measurements, but you've generated a lot of suggestive data that could be leveraged to push this field of research.*

Response: We apologize for not having provided a satisfactory response to this reviewer's question in response to our original submission. In L346-L358, we unpacked NDVI dynamics for different vegetation types (including grasslands and shrublands) under various meteorological conditions (e.g., different combinations of snow and monsoon precipitation). We also provided additional analysis that focused on identifying drought conditions between sites as well as fore-summer and post-monsoon droughts (see Line605-Line605 comment). We partly addressed the reviewer's comment about the role of snow precipitation and monsoon precipitation in ET and $R_{eco}$ dynamics using HPM estimations and we also emphasized the occurrence of energy limiting condition (L459-L469, also our response to Line604-Line605 comment). We understand the importance of splitting ET into evaporation/transpiration and $R_{eco}$ into heterotrophic and autotrophic respiration. However, additional datasets and laboratory environments (e.g., isotopes, water use efficiency data) would be needed for this, which is outside the scope of this study.

*2. Some of the results and discussion require more nuanced and/or focused interpretation (See detailed comments below). At the same time, the manuscript is long and could be shortened/tightened in many places to more accurately present/highlight key results (details below).*

Response: We have shortened the manuscript significantly (47 pages to 34 pages) and also worked to succinctly enhance nuanced interpretation. We also performed additional analysis based on the suggestions and comments. We hope we have struck a reasonable balance in this revision.

*L10: The decision to focus on Reco could be set up better. In other words, why Reco instead of NEE or GPP and/or all three? I don't necessarily have a problem with your decision to focus on Reco, but it must be clearly justified.*

Response: As is now described on L32-L36, $R_{eco}$ is sensitive to global climate change and plays a vital role in ecosystem carbon cycling (Le Quéré et al., 2009). Increases in $R_{eco}$ may contribute to a global warming acceleration through exerting positive feedback on the climate system (Cox et al., 2000). NEE and GPP are important, however better quantification and estimation of $R_{eco}$ is still needed in order to accurately quantify total carbon emissions from sparsely monitored ecosystems. This is the main reason why we focus on $R_{eco}$. In our response to reviewer #2's comments, we have developed specific HPM models to estimate NEE at certain FLUXNET sites and the model results are promising (Fig. A6). So we think HPM has the capability to provide NEE and GPP predictions and future studies may consider adopting our framework to better quantify net exchange of carbon and the assimilation component.

*L17: Sites within sites?*

Response: We improved clarity (L16).

*L21: Suggest adding "USA" here for the global audience.*

Response: We have modified correspondingly (L18).

*L27: Please specify "air, soil, snow", etc. whenever "temperature" is invoked. Lots of room for confusion here because most would expect ET to vary more with air temperature versus Reco that is more sensitive to soil temperature.*

Response: While we meant air temperature, we removed that sentence to shorten the abstract. In the revised manuscript, we have specified the use of 'air and soil' temperature to reduce confusion.

*L34-L35: Same comment as L10.*

Response: We have modified correspondingly

*L129-L131: Recent work by Chu et al. 2021 on the representativeness of statistical tower measurement footprints to surrounding areas may be relevant here.*

Response: We have described the work of Chu et al. (2021) in the revised manuscript (L50-L52).

L483: *Is it earlier snowmelt triggers the onset of vegetation activity or that higher air temperature trigger both snowmelt and the onset of vegetation activity?*

Response: In our study, we observed earlier increase of NDVI in years with earlier snowmelt (e.g., 2012) and later increase of NDVI in years with later snowmelt (e.g., 2015).
This observation is consistent with Pedersen et al. (2018). The relationship between NDVI, snowmelt timing and air temperature is non-linear in our study and thus we do not think it is higher air temperature trigger both snowmelt and the onset of vegetation activity. There are studies that reported a positive correlation between NDVI and temperature (Jia et al., 2006) but also no or even weakening relationship between vegetation activity and temperature variability (Piao et al., 2014). We did not intend to imply any causalities among these processes and we have made clarifications in the revised manuscript (L353-L355).

*L485-L486: Can you speak to the synoptic meteorological conditions in 2012 versus 2015? Why choose these two years for comparison? Similarly, the comparison of March, April and May between years is interesting, but what about the rest of the year? I'd be very interested in a similar post-monsoon analysis, potentially between years with strong and weak monsoons.*

Response: We chose year 2012 as it represents a severe fore-summer drought, and year 2015 because it was a normal/wet year based on the Palmer drought severity index (PDSI). This information has been added in L349-L352. In the revised manuscript, we have selected another year 2014, which was characterized by large snow precipitation but small monsoon precipitation. We added this year to better quantify dynamics for late-summer and autumn months (L390-L396). In addition to monsoon, we want to point out that there was a sharp decline in August (~30%) and September (~40%) radiation compared to June in the three years, indicating the potential of energy limiting condition rather than a monsoon moisture limiting condition (L465-L469). Figure 1 shows the distribution of incident shortwave radiation and similar trends are observed for net radiation that peaks in June (~ 180 $W\ m^{-2}$), and declines significantly in August (~ 90 $W\ m^{-2}$). Please also see our response to Line604-Line605 comment.

*L492-L497: Please edit this section to remove/acknowledge differences in NDVI that would be expected due to deciduous versus evergreen physiology. Some of the basic information currently comes across as results. I appreciate the attempt to relate these results back to processes, but this section needs refinement.*

Response: We modified the section correspondingly (L347-L349).

*L517: What does the "1" syntax correspond to?*

Response: For the East River sites, we selected 4 for each vegetation types. "1" is for the first one of each type as shown in Table 2. We clarified this in the manuscript.

*L525-L526: Please be specific about the meaning of "drought" in this context. Is it simply meant to connote some limitation to ET and/or Reco? If so, can you justify the underlying expectation that these variables would be affected at the same moisture threshold? I'd also argue that "usually" is the wrong word here. Earlier snowmelt certainly "can" trigger summer drought, but this scenario is subject to modification by monsoon precipitation and other factors as the authors acknowledge in this sentence. See recent work by Knowles et al. 2020, Xu et al. 2020, and many references therein.*

Response: We meant that earlier snowmelt is correlated with occurrences of fore-summer drought, and we agree with the reviewer that monsoon may modify drought conditions. We performed additional analysis to look deeper into drought conditions at the East River Watershed, please see our response to Line604-Line605 comment. Due to data availability, soil moisture was never used by HPM at the East River Watershed, so it is not feasible to expect how different soil moisture threshold influence ET and $R_{eco}$ predictions. We also want to point out that energy limiting condition for late-summer and autumn periods may occur as stated in our response to Line485-Line486 comment.

*L583-L596: I support this opportunity to discuss physiological differences between evergreen and deciduous vegetation, but simply citing Baldocchi et al. 2010 is insufficient. More thorough and nuanced discussion that incorporates foundational research on this topic is required.*

Response: Our original intent is to investigate whether HPM models can incorporate vegetation heterogeneity to quantify ET and $R_{eco}$ differences between different vegetation types with only 4 or 5 input features. We cited Baldocchi et al. 2010 to confirm that HPM estimation for deciduous forest and evergreen forest are reasonable and seek for physical explanation from their studies. This is mainly from a modeling perspective to explore limitation in model development and refinement; and a confirmation of model performance. We did not intend to characterize the physiology's control on ET and $R_{eco}$ as the only data we are currently using are meteorological reanalysis data and satellite data. We agree with the reviewer more thorough and nuanced research can advance our understanding of ecosystem dynamics, and we have added additional references that help us better understand the physiology's control on ET and $R_{eco}$ dynamics (L445-L451).

*L600-L601: See comment on L525-L526.*

Response: Please see our responses to Line525-Line526 and Line604-Line605 comments.

*L604-L605: This implies that growing season length determines snow water storage when in fact, it's closer to the opposite i.e., air temperature and/or snow accumulation determine the onset of the growing season. See Lian et al. 2020 and Zhang et al. 2020 for examples of more recent work on this topic. Combining the Sloat et al. 2015, Wainwright et al., 2020 and Hu et al., 2010 references here also raises an important distinction. Whereas the Sloat and Wainwright references invoke fore-summer i.e., pre-monsoon drought, the Hu reference pertains to late summer drought i.e., after snowmelt water inputs have subsided. This distinction reflects the typical relative importance of snowmelt vs. monsoon precipitation at a given site e.g., snow-dominated sites may be susceptible to moisture limitation after the snowmelt pulse (late summer; Hu et al. 2010), whereas monsoon-dominated sites may be susceptible to moisture limitation before the onset of monsoon rains (early/fore-summer; Sloat et al. 2015; Wainwright et al. 2020). Please establish the typical relative importance of snow versus monsoon precipitation at the East River site and how your results may be expected to change at sites where moisture availability is typically more or less affected by snowmelt versus monsoon precipitation.*

Response: We agree that L604-L605 was misleading. We have clarified the sentence. In addition, we clarified the typical relative importance of snow versus monsoon precipitation on ET at East River site in the revised manuscript (L380-L396).

With regard to the studies the reviewer is referring, we note that Sloat et al. (2015) used peak net ecosystem productivity and Wainwright et al. (2020) used peak June NDVI as measures for fore-summer periods at the East River sites whereas Hu et al. (2010) used annual carbon uptake and growing season length at Niwot Ridge. Though they have chosen different metrics in their studies, we do not think there's a distinct difference at Niwot Ridge (US-NR1) or East River that one site is more snow-dominated versus monsoon-dominated, or that one site constrained by fore-summer drought or post-monsoon drought. Here we used SNOTEL Butte (ER-BT) as a representative site for the East River Watershed due to data availability.

In fact, US-NR1 and the East River watershed share lots of similarities (e.g., in the same ecoregion). Precipitation, temperature and elevation are similar for US-NR1 and ER-BT (Table 1). Palmer drought index (PDI) and Palmer drought sensitivity index (PDSI) were used to quantify drought conditions, as documented in Sloat et al. (2015) and Wainwright et al. (2020). We did not find any quantitative measures for droughts in Hu et al. 2010. None of these three studies derived any indices to explicitly quantify post-monsoon drought conditions, so we used August PDSI to compare them. Figure 1 presents the PDSI time series obtained from Abatzoglou et al. (2018) for US-NR1 and ER-BT. Based on the U.S. drought monitor classification, a value of $-1$ is the threshold for droughts. And the more negative PDSI values are, the more severe the droughts are. If PDSI values are greater than $-1$, the ecosystems may not experience drought condition.

**PDSI time series**

[Figure]

Figure 1. Time series of PDSI at ER-BT and US-NR1. Values smaller than -1 indicate drought condition.

We applied a simple linear regression of these PDSI values between US-NR1 (Hu et al. 2010) and ER-BT (Wainwright et al. 2020). We found a correlation coefficient of 0.88 (p < 2.2e-16), 0.82 (p < 2.2e-16) and 0.91 (p < 2.2e-16) for annual, June and August mean PDSI values between the sites, respectively. PDSI values in 2008 and 2014 differ significantly between the two sites, however that was mainly caused by unusual precipitation events and outside period with drought conditions as PDSI is greater than −1. Based on this result, we believe it is reasonable to conclude that the drought conditions for US-NR1 and East River Watershed are similar.

[Figure]

Figure 2. Net radiation distribution from 2011 to 2016 grouped by month at the East River Watershed.

We also discovered a high correlation between June PDSI and August PDSI. The correlation coefficients are 0.98 (p < 2.2e-16) and 0.90 (p < 2.2e-16) for US-NR1 and ER-BT, respectively, which indicates the coherency of fore-summer drought and post-monsoon drought if any. We want to note that PDSI has its own limitations, and we were not able to explore other data products that may be more sensitive to monsoon precipitation. Still, this result indicates occurrence of post-monsoon droughts are highly correlated with the occurrence of fore-summer droughts. Individual monsoon events may change the soil moisture condition in short terms, however may not entirely alter the drought conditions. We also want to point out to a recent work by Carroll et al. (2020), where they discovered July-September monsoon in the central Rocky Mountains may support ET in lower subalpine forests, but the monsoon precipitation also contributes to streamflow deficiencies caused by reductions in snow accumulation. They suggested that the timing and location of water input with respect to energy and water availability remain key issues. If monsoon arrives when potential ET (PET) is high and soil moisture is waning during fore-summer droughts, this water serves to moisten dry soils and is consumed very quickly by vegetation leading to increases in ET (moisture limiting condition). But if the timing of monsoon arrives late when PET is small, monsoon precipitation may contribute to streamflow rather than ET as the ecosystem is under energy limiting condition. In our study, we observed a significant decline in radiation after peak growing season regardless of the amount of monsoon precipitation. Net radiation declines by ~ 30% in August and ~ 40% in September compared to June. During the late-summer and autumn months, we think the East River Watershed is more likely to be constrained by energy rather than moisture limitation during late-summer and autumn months. We provide revised text at L378-L396; L456-L472.

*L612: Hard to follow, I think "whereas" may be the wrong word here.*

Response: We meant to say that earlier arrival (early-July in 2012) of monsoon precipitation help buffer the fore-summer drought condition. Correspondingly, 2012 July ET is not substantially different compared to other years.

*L629: "Microclimate" is misspelled.*

Response: We removed this sentence in the revised manuscript to shorten the manuscript. We have made sure spelling is accurate throughout the manuscript.

**Our response to RC#3**
**Authors' response to RC3 review**
We appreciate the anonymous reviewer for reviewing our manuscript and provide constructive for us to better improve the manuscript.
Major remarks:
*1. I read parts of the manuscript several times to understand how the FLUXNET and CLM data was used (, separately) and how the framework exactly works, and I am still not sure if I entirely understand it. Also, it took me some time to understand the four experiments ("use cases"), what data was used for training, testing, etc. This is my major critic: I think the manuscript needs a cleaner structure and language.*

Response: In the revised manuscript, we have added a section to demonstrate the four use cases and indicate the relevant data used for training and validation (L271-L284).

*2. For me, the term "hybrid" is a bit confusing here. I assume that you refer to Reichstein (2019), where "(5) Surrogate modeling or emulation" is listed as a hybrid approach, which, once trained, can "achieve **simulations orders of magnitude faster than the original physical model without sacrificing much accuracy**" and "allows for fast sensitivity analysis, model parameter calibration, and derivation of confidence intervals for the estimates". I think the manuscript would be much easier to understand if you would make this clearer.*

Response: We use 'hybrid' in HPM to indicate the use of machine learning with mechanistic-based models/output and FLUXNET measurements integrating with other datasets, such as remote sensing. We show how HPM approach was used to 1) couple flux measurement for gap filling and time series prediction (Use case 1); 2) integrate flux measurement for spatial reconstruction and configuration in different ecoregions (Use case 2); 3) implement with physical process models (Use case 3) and 4) provide flux estimation to gain better understanding of ecosystem dynamics (Use case 4).. We have better clarified these points in the revised manuscript (L12-L14; L115-L118).

Minor Remarks:
General:
*I strongly recommend to use colorblind-friendly colors in the plots. The time-series plots with green and red color mixed are particularly problematic. I think that the figures need some more work (general appearance, font size).*

Response: We have made the necessary changes.

*From the HESS guidelines: "Common Latin phrases are not italicized (for example, et a;., cf., e.g., a priori, in situ, [...])" (e.g., line 49, in situ).*

Response: We have made the necessary changes.

*From the HESS guidelines: "The abbreviation "Fig." should be used when it appears in running text and should be followed by a number unless it comes at the beginning of a sentence, e.g.: "The results are depicted in Fig. 5. Figure 9 reveals that ...".".*

Response: We have made the necessary changes.

*From the HESS guidelines: "Units must be written exponentially (e.g. $W\ m^{-2}$)." e.g., line 380 or in axes labels, you use mm/d instead of m d$^{-1}$.*

Response: We have made the necessary changes.

*You use the notation "Adj.R2-0.94" in some figures (e.g. Fig. 5). This is misleading, please use "Adj.R2: 0.94", "Adj.R2=0.94", or similar.*

Response: We have made the necessary modifications.

*Time-series figures: please add a legend for all plots (pink points, red, green, blue, black lines).*

Response: We have made the necessary modifications.

*Symbol notation: I noticed you use "ET" for evapotranspiration and "$R_{ECO}$" for ecosystem respiration. I find this is inconsistent, as you either you use these as abbreviations, which are not italic ("ET" & "$R_{ECO}$"), or as mathematical variables ("E"*

& "$R_{ECO}$"), where multi-letter symbols are to be avoided due to ambiguity (is "$ET = E \cdot T$?), and subscripts are only italic if they refer to a variable (such as in $x_i$, where $i$ is an index), but not if the subscript is a name.

Response: Thank you for the comment. We have made the necessary changes.

*In general, many small "not so nice" things like units written inconsistently.*

Response: We appreciate the comments and have made necessary changes.

*I suggest to not put "learn" in quotes (as in the model "learns") as the term is very commonly used in this context.*

Response: We have made the necessary modifications.

*Nice that you split the data in training, validation, and test (prediction) set! This is often not done.*

Response: Thank you

*The abstract is too detailed in my opinion, consider to shorten.*

Response: We have made the necessary modifications (32 lines to 22 lines).

*I suggest to state clearly how the approach is hybrid and why you use the approach.*

Response: We have increased the clarity (L115-L118).

*Nice review of current methods to estimate ET and $R_{ECO}$. It could be shortened a bit.*

Response: We have made the necessary modifications (24 lines to 16 lines).

*Tab.1 It is hard to differentiate between the rows visually.*

Response: We have made the necessary changes (L154).

*Fig.1 Consider highlighting the SNOTEL sites visually.*

Response: We used different shapes and colors to distinguish different sites (L157).

*I think you don't need to explain the LSTM in detail.*

Response: We have made the necessary changes (L209-L221).

*L260: Does "deeply connected neural networks" refer to a fully connected neural network?*

Response: Yes.

*For use case 2, do you train the model on all sites jointly or on single sites?*

Response: We trained the model on individual sites.
*L282-L320: Consider replacing the extensive description of LSTMs with a conceptual high-level description.*

Response: We have made the necessary changes (L209-L221).

*L326: Would be nice to see if a smaller model does the job (but not essential here).*

Response: The current configuration of neural networks does not require any super-computing power and we were satisfied with the prediction accuracy.

*L331: Olah. (2015) -> Olah (2015)*

Response: We have made the necessary changes (L221).

*L340-L352: Why did you separate precipitation into rainfall and snowfall and how was the variable sn used? If they were used as inputs for the LSTM, why not letting the neural network figure this out, i.e., just inputting the available features?*

Response: At locations dominated by snow, timing of snowmelt and bareground date is important for ET and $R_{eco}$ dynamics. As there are only 4 or 5 features currently used, manual separation of precipitation into rain and snow may help the model establish linkages between precipitation and energy perspectives to better learn ecological memories and thus improve model performance. At locations where snow is rarely present, precipitation was directly used. We clarified this in the paper.

*L355: I assume you used an LSTM? Then you can just use the term LSTM here, as it has been introduced already instead of "deep-learning recurrent neural networks".*

Response: Yes. We have made the necessary changes (L264).

*I suggest to move the descriptions of the "use cases" to the methods section, maybe make a table that summarized what data is used for training and testing, the objective of the experiment etc.*

Response: We have made the necessary changes. A new section has been added (L271-L284).

*The interpretation would be much easier if you would show the mean seasonal cycle and the interannual variability!*

Response: Thank you for your comment. We intended to use the monthly mean comparisons to show seasonal cycles and interannual variability. In discussion sections, we provided more details about ET and $R_{eco}$ at specific years.

*L399-L407: This is already discussion of the results.*

Response: We have made the necessary changes.

*L399-L404: I would expect that the LSTM learns SM dynamics i.e., it represents it (implicitly) in its hidden state. SM would not necessarily be needed as the LSTM earns the ecological memory effects (e.g., Besnard et al. (2019) or Kraft et al. (2019). Adding SM could still help improving the model as it currently does not have much data to learn from compared to the number of parameters. Also, referring to a comment from former Referee #1, I think this should be clarified. This is one of the key advantages of using models like an LSTM, it can learn ecological memory and thus, variables such as soil moisture may not be needed!*

Response: We agree with the reviewer that LSTM does has the advantage it could in theory learn the ecological memory. Still, we have to recognize that results of this study show that the use of LSTM cannot replace entirely the information present in soil moisture. Results show that ET and $R_{eco}$ estimations at sites limited by energy condition have very high estimation accuracy, which suggests LSTM was able to capture the ecological memories. However, at sites that experiences drought conditions, some of ET and $R_{eco}$ anomalous values are not frequent enough for LSTM to learn. These are time period where soil moisture data can be useful for this case to better inform LSTM and further increase prediction accuracy.

*L405-L407: I agree that LSTMs tend to have issues with extreme values. In my opinion, this is mostly because extreme values are rare, i.e., the model does not see many anomalous samples, there is less training data for such cases. Maybe you could mention this and provide a source, if you can find one.*

Response. We agree with the reviewer. We have elaborated on this issue (L492-L497).

*Tab. 3: Please write units in exponential form. You could mention that the increase in test performance could be linked to dropout (which I assume is deactivated for inference) in the discussion.*

Response. We have made the necessary changes.

*I think the representativity of FLUXNET sites for the entire ecoregion is questionable and disputed (?), maybe rephrase.*

Response. We have elaborated on this (L306-L308).

*Fig. 7: The monthly errors used to be pink before, right? I suggest to reuse the same colors.*

Response. In the revision, we have adopted a consistent color scheme and palette.

*L450: I don't know what an "1-D" model is, consider explaining.*

Response. We were referring to the 1-dimensional CLM model developed in Tran et al. It solves physical equations in the vertical direction (L324-L326).

*L475: The mechanistic HPM model?*

Response. We have made the necessary changes (L341).

*L479: 30m -> 30 m.*

Response. We specified the resolution of remote sensing data in L82 and removed this sentence to shorten the manuscript.

*L479+ Much of it is discussion.*

Response. We have made the necessary changes.

*L517: 17% -> 17 %*

Response. We have made the necessary changes (L372).

*Fig. 11: Panels (a) and (b) are not very informative, maybe remove?*

Response. We think panels (a) and (b) are needed as they show the temporal trends and explain the seasonality of ET and Reco estimation at the East River Watershed for deciduous forest. Panels (a) and (b) placed the background for the following panels. Thus we decided to keep these two panels in Fig. 11

*L559: You referred to "physically-model-based HPM" as "mechanistic HPM" (line 264), you may use the latter one here.*

Response. We removed this sentence during revision to shorten the paragraph. But yes, it should be 'mechanistic HPM'.

*L625: Again, I think you need to discuss the "memory aspect". If you have meteorological data and site-level variables (e.g. vegetation type, soil properties), and enough training data, an LSTM would learn SM implicitly. This should be added to the discussion, as it is a key selling-point for using deep learning models. I think the message "SM is needed for improving model" is wrong, state variables are not needed anymore with DL approaches if the states can be derived from the input data. Of course, it can still*

*be beneficial to add soil moisture, as it would regularize the model and maybe, the complex processes involved (e.g., lateral flux) may not be learnt by the model if the relevant features are missing.*

Response. We agree with the reviewer that LSTM has been successful capturing the ecological memory effects in our study as well, and we have acknowledged this perspective in the revised manuscript (L426-L429). However, our results at certain sites suggest that drought occurrence and moisture limiting conditions may not be well captured by LSTM. We agree with the reviewer that soil moisture should be derived from the input data, but challenges still remain. There are uncertainties in the meteorological inputs (L405-L411), which increases the difficulties for LSTMs to learn soil moisture implicitly. LSTM may not be sufficiently trained upon drought conditions and longer time series may improve model performance. Soil moisture data can potentially fill the gap between atmospheric forcings and site-specific information. Thus at the current stage, we recommend to include soil moisture data when available to bypass certain limitations in data inputs and insufficient training. We have increased the clarity in the revised manuscript (L475-L481).

*L651-L660: As an outlook: the model could be trained on FLUXNET and process-based simulations jointly.*

Response. We have elaborated on this point (L495-L497).

*L669: I cite reviewer #1: "Replace CO with Colorado, USA for the global audience."*

Response. We have made the necessary changes (L506).

[revised manuscript text omitted]
). June air temperature differences among SNOTEL sites were occasionally over 3 ℃ , while the DAYMET data from the East River rarely revealed 0.2 ℃ differences. In addition, a ~80 $W$  $m^{-2}$ of radiation differences was observed with SNOTEL data whereas radiation differences stays around 30 $W$  $m^{-2}$ for East River sites.

Correspondingly, we observed 2.5 times greater differences in ET across SNOTEL stations compared to the sites within the East River watershed. We observed similar level of differences (around 0.8 $gCm^{-2}$) in $R_{ECO}$ $R_{eco}$ within

East River Watershed and across SNOTEL stations. Landsat data enabled us to capture NDVI differences at these sites, but we have identified the insufficient resolution of input meteorological forcing data at the East River sites.

These results indicate uncertainties in meteorological forcing attributes (e.g., radiation and air temperature) can have a huge influence over HPM ET estimation and HPM $R_{eco}$ model is more sensitive to

[revised manuscript text omitted]

---

## Author Response (AR3)

We thank the editor and two anonymous referees for reviewing our manuscript and providing us with constructive feedback. Below are our point-by-point responses to the referees' comments, followed by a revised manuscript showing all track changes made through the minor revision. Note referee's original comments are in italic and all line numbers are based on the revised manuscript

Reviewer #1:

*General Comment: I appreciate the thorough response to my previous comments, especially with respect to discussion of pre-monsoon versus post-monsoon drought. I understand this is a nuanced distinction and that your model was not designed to identify snow- and rain-mediated differences in ecosystem fluxes. I have only a few remaining minor comments.*

*It's still unclear to me why the study considers ET and Reco but not GPP and/or NEE. i.e., two of the four main ecosystem carbon and water fluxes are included and the other two are excluded. What was the reason for this? I don't mean to be picky, and I'm certainly not suggesting that you re-do any analyses to include additional data, but readers will be wondering the same. Adding a sentence or two to justify/explain your rationale for your choice in the introduction, preferably early on, would go a long way toward setting appropriate expectations.*

Response: We appreciate the reviewer for the summary of our work and feedbacks of our manuscript. We agree with the reviewer that GPP and NEE are important fluxes. In our previous response to reviewer comments, we developed HPM models to estimate NEE at several FLUXNET sites (Fig. A6). Based on those promising results, we think our proposed HPM framework has the capability to estimate NEE and GPP. In the revised manuscript, we have justified the rationale for our choice of $R_{eco}$ (L26, L32-L37).

*Table 1: You might consider adding a "Site name" column that would be helpful to readers that are familiar with these sites e.g., "Niwot Ridge" or similar for US-NR1. The units for precipitation are mislabeled as "(m)"; precipitation values could be rounded to the ones place.*

Response: We have added a 'Site Name' column in Table 1 and corrected the unit for precipitation.

*Line157: Change "closed" to close.*

Response: We have made the necessary correction (L157).

*Line401-417: This paragraph in particular would benefit from a grammar edit.*

Response: We have edited the paragraph thoroughly and made necessary changes (L401-L415)

Reviewer #2:

*General comment: This study uses a novel approach that combines process-based modeling and eddy covariance measurements with freely-available spatial datasets and machine learning to predict evaporation and respiration fluxes at the ecosystem scale. The input training data are meteorological data and remote sensing NDVI, which are linked to Measured (via eddy covariance, aka 'data HPM') or modeled (via Community Land Model, aka 'mechanistic HPM') DAILY et AND ECOSYSTEM RESPIRATION. The method used is a long short-term memory (LSTM) model – which is a type of recurrent neural network that has been used successfully for rainfall runoff and hydrological monitoring modeling. The authors demonstrated the use of both a data HPM and mechanistic HPM approach, and then used the latter method to model ET and Reco at several locations within the East River Watershed in southwestern*

*Colorado. The authors then relate the resulting modeled fluxes to changes in biome and year-to-year weather patterns. They conclude that HPM is a feasible method for estimating spatial and temporal patterns of ET and Reco, and I agree with them. This paper is written logically, with very few typos and needs just a bit of clarification (see my comments below). This paper is an interesting study on using data and tools available to us to address the need for spatially- and temporally-dense water and carbon fluxes, especially in mountainous landscapes. I am not an expert in neural network modeling, but I found the methods were not too difficult to follow.*

Response: We thank the reviewer for the high-level summary of our work and positive remarks of our manuscript.

Line-by-line comments:

*Line111: 'multi-model and multi-data approaches…'*

Response: We have made the necessary change (L113)

*Line223: Shouldn't 'CLM predictions' be added to the list of input features?*

Response: We have made the necessary change (L223)

*Line263: It is not clear to me why reconstruction of daily NDVI is necessary. Why can't the NDVI measurements be used with the gaps in time coverage? As long as there are accompanying meteorological measurements for that day, it should be fine. Please briefly add a statement justifying why it was necessary to gap-fill NDVI data and how many data points and the percentage of total NDVI data points were modeled instead of measured.*

Response: Our current HPM configuration requires all input data to have the same temporal scale (i.e., daily), which is the main reason that we reconstruct daily NDVI time series (L264-L265). Thus, the NDVI reconstruction step is still necessary for HPM. Around 10% of the NDVI data points were measured given the 16-day revisit time of Landsat 8, which can be improved with ongoing advances in remote sensing.

*Line501: 'These models were trained using and validated against eddy covariance…'*

Response: We have made the necessary correction (L499).

[revised manuscript text omitted]